# Conformation of the nuclear pore in living cells is modulated by transport state

Joan Pulupa[1], Harriet Prior[1], Daniel S Johnson[2], Sanford M Simon[1]*

[1]Laboratory of Cellular Biophysics, Rockefeller University, New York, United States; [2]Department of Physics and Astronomy, Hofstra University, Hempstead, United States

**Abstract** While the static structure of the nuclear pore complex (NPC) continues to be refined with cryo-EM and x-ray crystallography, *in vivo* conformational changes of the NPC remain underexplored. We developed sensors that report on the orientation of NPC components by rigidly conjugating mEGFP to different NPC proteins. Our studies show conformational changes to select domains of nucleoporins (Nups) within the inner ring (Nup54, Nup58, Nup62) when transport through the NPC is perturbed and no conformational changes to Nups elsewhere in the NPC. Our results suggest that select components of the NPC are flexible and undergo conformational changes upon engaging with cargo.

## Introduction

Recent advances in structural biology have allowed the characterization of the static structures of large macromolecular complexes. However, describing the conformational changes of proteins within these complexes, especially when the proteins are in their native cellular context, has proven challenging. In this paper, we establish a technique to visualize the orientations of domains of proteins *in vivo* and we apply this technique to study the conformational changes in the nuclear pore complex (NPC).

The scaffold of the NPC is a cylindrical channel composed of proteins that form an eight-spoked core with an axis perpendicular to the nuclear envelope (*Alber et al., 2007*; *Kim et al., 2018*; *Kosinski et al., 2016*; *Mosalaganti et al., 2018*). NPCs are composed of ~30 distinct proteins called nucleoporins (Nups) represented in 8, 16, or 32-fold copy number. An individual yeast NPC is composed of ~500 Nups for a total mass of ~66 MDa (*Rout and Blobel, 1993*), whereas a vertebrate NPC has ~1000 total Nups (*Hoelz et al., 2011*) fpr a tptal mass of ~109 MDa (*Reichelt et al., 1990*). From the scaffold of the NPC, relatively unstructured domains of Nups protrude into the lumen of the cylinder. These are known as 'FG-nups' because they contain repeat motifs of phenylalanine-glycine that interact with cargo and its chaperones, known as the importins, exportins, or karyopherins (kaps). While the organizing principles of the NPC are shared across eukaryotes, there are notable variations even between human cell lines. Differential expression levels (*D'Angelo et al., 2012*; *Lupu et al., 2008*) and stoichiometries (*Ori et al., 2013*) of Nups are observed between different cell types. Furthermore, within the mammalian NPC, different Nups have been shown to have different residence times, which has led to the suggestion that those with shorter residence times may serve adaptor or regulatory functions (*Rabut et al., 2004*). The additional mass of the mammalian NPC exists in the absence of substantial changes to the size of the central channel, raising the possibility that the mammalian NPC is subject to additional regulation.

In the adherent cell lines used in this study, the nuclei tend to be flattened ovoids and the NPCs on the basal surface share a common orientation with their central axis perpendicular to the coverslip. Since each component of the NPC is in 8-fold symmetry, it is a compelling test structure to assay conformational changes *in vivo*. Using polarized-total internal reflection

*For correspondence:
Sanford.Simon@rockefeller.edu

Competing interests: The authors declare that no competing interests exist.

fluorescence microscopy (pol-TIRFM), we monitored the orientation of mEGFP-based sensors incorporated into different domains of individual NPCs in living cells. Previously, to measure the organization of various Nups with respect to the NPC we used fluorescence anisotropy. We determined the organization of the Y-shaped complex with respect to the NPC (*Kampmann et al., 2011*) and characterized the orientations and rigidity of the FG-Nup domains (*Atkinson et al., 2013*; *Mattheyses et al., 2010*). The anisotropy approach used polarized light to excite many dozens of NPCs and then measure the emission parallel and perpendicular to the excitation.

In this study, we examine the orientation of components of the scaffold and establish that the conformation of the inner ring of the NPC is modulated by both transport state and specific transport factors. To monitor conformational changes in the scaffold of the NPC, we built orientational sensors by rigidly attaching mEGFP to different Nup domains. We conjugated the alpha helix at the amino terminus of mEGFP to the carboxyl terminus of a Nup domain. Thus, the orientation of mEGFP is fixed to that of the Nup. The orientation of mEGFP can be monitored because the excitation dipole is fixed within the molecule. The strength of mEGFP excitation is proportional to $\cos^2(\Theta)$, where $\Theta$ is the angle between the excitation light and the excitation dipole. To monitor the orientation of the mEGFP, we excited the fluorophore sequentially with light polarized in two orthogonal directions. To restrict excitation to the basal surface of the cell we created the two polarized fields using excitation by total internal reflection (TIR). Because the basal nuclear envelope is parallel to our coverslip, the NPCs embedded in the nuclear envelope are oriented relative to the optical axis of our microscope. Therefore, we could monitor the orientation of Nups within individual NPCs in living cells. Our results show a rearrangement of the structural core of the NPC, in particular of the inner ring Nups, in response to manipulations of transport through the NPC.

## Results

### The orientation of Nup-mEGFP fusion proteins can be monitored with polarized-total internal reflection fluorescence microscopy (pol-TIRFM)

We alternated the polarization of the excitation field between $\hat{p}$-polarized light (perpendicular to the coverslip and parallel to the nucleo-cytoplasmic axis of each NPC) and $\hat{s}$-polarized light (parallel to the coverslip and perpendicular to the nucleo-cytoplasmic axis of each NPC). We then measured the light emitted from each Nup orientational sensor in response to the two orthogonal polarizations of TIRF illumination (*Figure 1A*).

We probed for conformational changes in the NPC with orientational sensors in Nup133, Nup93, Nup54, and Nup58. Nup133 is a member of the Y-shaped complex, which forms two reticulated rings at both the nuclear and cytoplasmic faces of the NPC (*Bui et al., 2013*). Nup93 is a member of the Nup93 complex (along with Nup205, Nup188, Nup155, and Nup53) and Nup54 and Nup58 are members of the Nup62 complex (along with Nup62). Both the Nup93 and Nup62 complexes localize to the inner ring (*Vollmer and Antonin, 2014*), between the two reticulated rings of Y-shaped complexes. The Nup93 complex is embedded within the NPC scaffold, and the Nup62 complex sits adjacent, closer to the lumen of the nuclear pore (*Kosinski et al., 2016*). Therefore, our orientational sensors were localized to three different structural positions within the NPC (*Figure 1C*).

Plasmids encoding putative orientational sensors were engineered to encode each Nup fused via an alpha helix to mEGFP. We removed the first four amino acids of the mEGFP because they are not resolved in the crystal structure, suggesting that they are not rigid. We transfected these plasmids into HeLa cells (*Figure 1A*, *Figure 1—figure supplement 1*). With an algorithm that uses a Laplacian of blob-detection method (*Pulupa, 2020*) the distribution of total intensities of the puncta observed was only a single peak indicating we are detecting individual NPCs. The algorithm extracted light emitted in response to $\hat{p}$ and $\hat{s}$ excitation to calculate what we refer to as the p:s ratio (*Figure 1B*, *Figure 1—figure supplement 1*). If the mEGFP is rigidly conjugated to a Nup that is properly incorporated into the NPC, then changes of the p:s ratio represent changes in orientation of the domain of the Nup to which the mEGFP is conjugated.

If the mEGFP is held rigidly relative to a Nup, then changing the length of the alpha-helical linker by a single amino acid should rotate the mEGFP ~103° around the axis of the alpha helix. Therefore, if our reporter is a proxy for the orientation of the Nup, the p:s ratio should shift with the number of amino acids in the linker. This hypothesis was tested by inserting rigid or flexible linkers of different

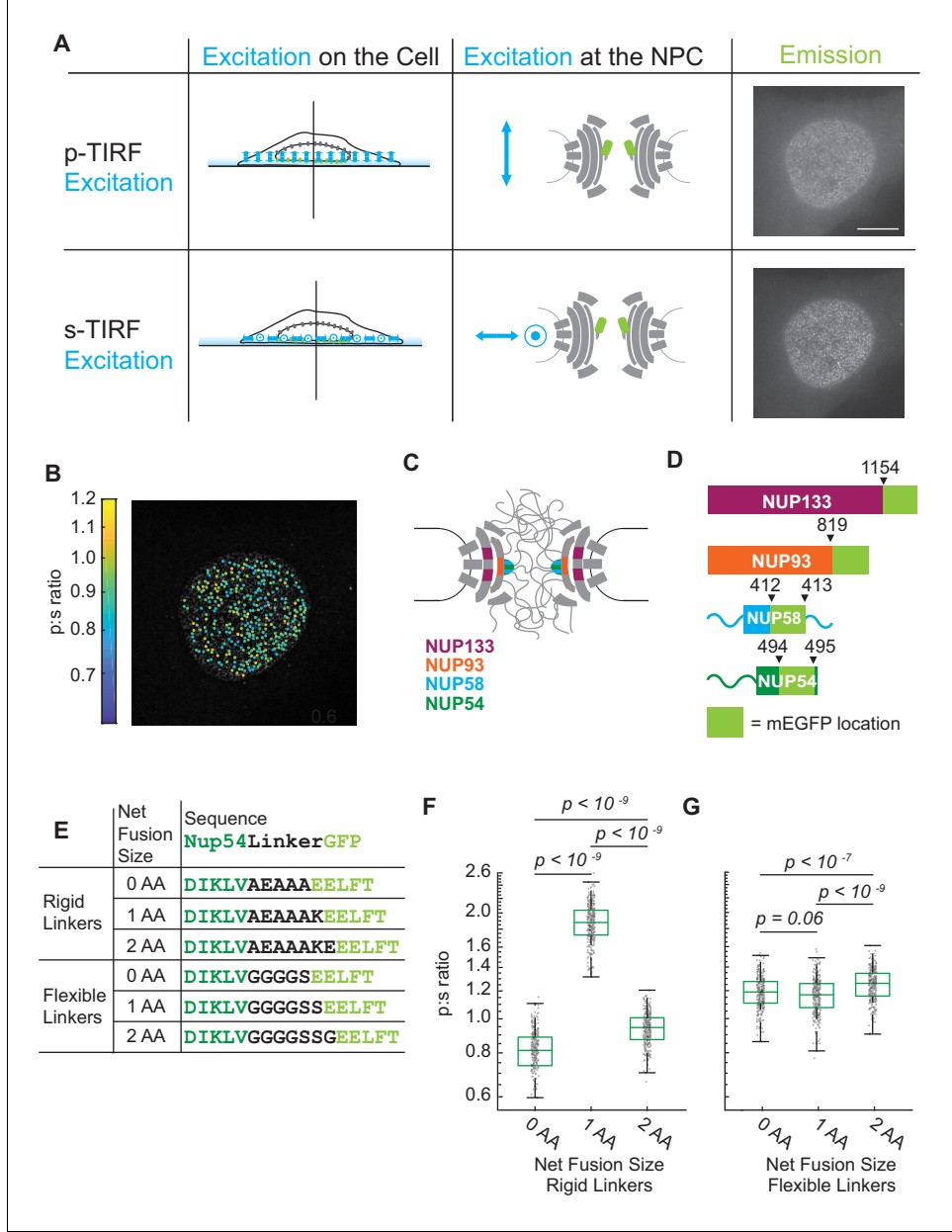

**Figure 1.** The orientation of Nup-mEGFP fusion proteins can be measured in individual NPCs with pol-TIRFM. (**A**) Using pol-TIRFM, the bottom of the nucleus is illuminated and Nup-mEGFP fusion proteins are excited with $\hat{p}$-polarized or $\hat{s}$-polarized light. $\hat{p}$-excitation is parallel and $\hat{s}$-excitation is perpendicular to the nucleocytoplasmic axis of the NPC. The emission from Nup54-mEGFP fusion proteins in HeLa cells in response to each excitation is shown in the right column. (Scale bar = 10 μm). (**B**) p:s ratios, a measurement of orientation of mEGFP, are calculated for each NPC and represented with a color scale. (**C**) Schematic of the NPC indicating the Nups studied. (**D**) Schematic of the mEGFP placement within the Nups. (**E**) Nup54-mEGFP constructs with flexible or rigid linkers in between the Nup and the mEGFP. With each additional amino acid in a rigid alpha helix, the mEGFP rotates 103° relative to the Nup, which does not happen with a flexible linker. (**F**) The p:s ratios of Nup54-mEGFP[494] fusion proteins with a rigid linker shift with the addition of each amino acid. (**G**) The p:s ratios of Nup54-mEGFP[494] fusion proteins with a flexible linker do not shift upon amino acid additions. (n = 300 NPCs, 10 cells, boxes indicate quartiles, center bars indicate medians, one-way ANOVA with post-hoc Tukey test).

The online version of this article includes the following source data and figure supplement(s) for figure 1:

**Source data 1.** Source data for *Figure 1*.

**Figure supplement 1.** The orientation of Nup-mEGFP fusion proteins can be monitored with pol-TIRFM.

**Figure supplement 1—source data 1.** Source data for *Figure 1—figure supplement 1*.

lengths between mEGFP and the terminal structured alpha helix of Nup54 (*Figure 1E*), which is defined as residues 456–494 (*Solmaz et al., 2011*). The constructs were transiently transfected into HeLa cells, and the p:s ratios were measured in living cells 24–48 hr post transfection. The p:s ratio shifted with different lengths of the rigid linkers, confirming that the dipole of the mEGFP is a proxy for the orientation of the Nup54 (*Figure 1F*). No difference in p:s ratio was observed with varying lengths of the flexible linkers (*Figure 1G*). Thus, we developed a criterion whereby orientational sensors are considered functional if the p:s ratio shifts upon the addition of a single amino acid.

Using this criterion, we confirmed that we have orientational sensors with mEGFP conjugated to Nup133, Nup93, Nup58, and Nup54 (*Figure 2A–D*). We saw distinct shifts in the p:s ratio when constructs differed by a single amino acid. An exception was when mEGFP was placed at the carboxyl terminus of Nup54. This domain is currently unresolved in any crystal structure, and the p:s ratio did not change upon rotating the linker alpha helix (*Figure 2E*). This indicates that this domain of Nup54 is not held rigidly, consistent with the inability to crystallize this domain of the protein and suggests the carboxyl terminus is a flexible domain. For any specific Nup-mEGFP, variations in the p:s ratio in different NPCs were not detected with respect to position along the basal surface of the nucleus (*Figure 1—figure supplement 1*), indicating a shared orientation for these NPCs.

## The orientations of inner ring nups change after starvation

In yeast, starvation inhibits the import of cargo with nuclear localization sequences, or NLS-tagged cargo (*Stochaj et al., 2000*). We tested the effects of starvation on the orientation of Nups. Cells were starved for 24 hr in Hank's Balanced Salt Solution (HBSS). We confirmed the rates of nucleocytoplasmic trafficking were attenuated using a photoactivatable nuclear transport cargo (*Yumerefendi et al., 2015*; *Figure 3—figure supplement 1*).

After starvation there were no detectable changes in the orientations of Nup133 and Nup93 (*Figure 3A–B*, *Figure 3—figure supplement 2A–D*). These results are consistent with the NPC retaining its orientation relative to the coverslip. The consistency across the basal surface of the nucleus shows the nuclear envelope is not distorted post-starvation.

In contrast, conformational reporters Nup58 and amino acid 494 in Nup54 (Nup54-mEGFP[494], or Nup54[494]) exhibited significant shifts post-starvation (*Figure 3C–D*, *Figure 3—figure supplement 2E–H*). These orientational shifts in select alpha helices of Nup54 and Nup58 reflected a reorganization of these inner ring Nups relative to the NPC. The orientation shift of Nup54[494] was also seen in Nup54 orientational sensors where the FG-Nup domain was eliminated (*Figure 3—figure supplement 3*). This result suggests that these orientational changes are propagated throughout the inner ring, independent of whether the individual Nup54 polypeptide containing the orientational sensor is bound to a kap. No shift was observed in the previously described reporter at amino acid 510 in Nup54 (Nup54-mEGFP[510], or Nup54[510]), the carboxyl terminus (*Figure 3E*). These results suggest a change of orientation of select alpha-helical domains of Nup58 and Nup54 in the absence of a change in the orientation of the rest of the NPC.

## Nup-mEGFP orientational reporters are functional

To ensure that the Nup-mEGFP fusion proteins are functional and to improve the signal to noise, we used CRISPR/Cas9 to create cell lines in which both endogenous copies of either Nup133 or Nup54 are replaced with their fluorescent orientational sensor equivalent.

We confirmed that these cell lines are homozygous (*Figure 4—figure supplement 1*) and observed no changes in cell growth or morphology (*Figure 4A–B*). These cell lines also shared the property that single amino acid alterations in the linker length shifted the p:s ratio (*Figure 4C–D*). The Nup133 and Nup54 CRISPR cell lines exhibit similar p:s ratios to the transient transfections. However, we measure a difference in the p:s ratio of a little above one in the Nup133 CRISPR cell lines and below one in the transient transfections. This difference may reflect a slight alteration in NPC structure between two different cell types or is a methodological consequence of either variable incorporation levels of fluorescent Nup133 in the transient transfection or the higher background fluorescence in the transient transfection, which might slightly lower the ratio from above 1 to below 1.

When starved, the CRISPR cell lines mimicked the results of the transiently transfected cell lines (*Figure 4E–F*, *Figure 4—figure supplement 2*); the orientation of Nup54[494] exhibited a significant

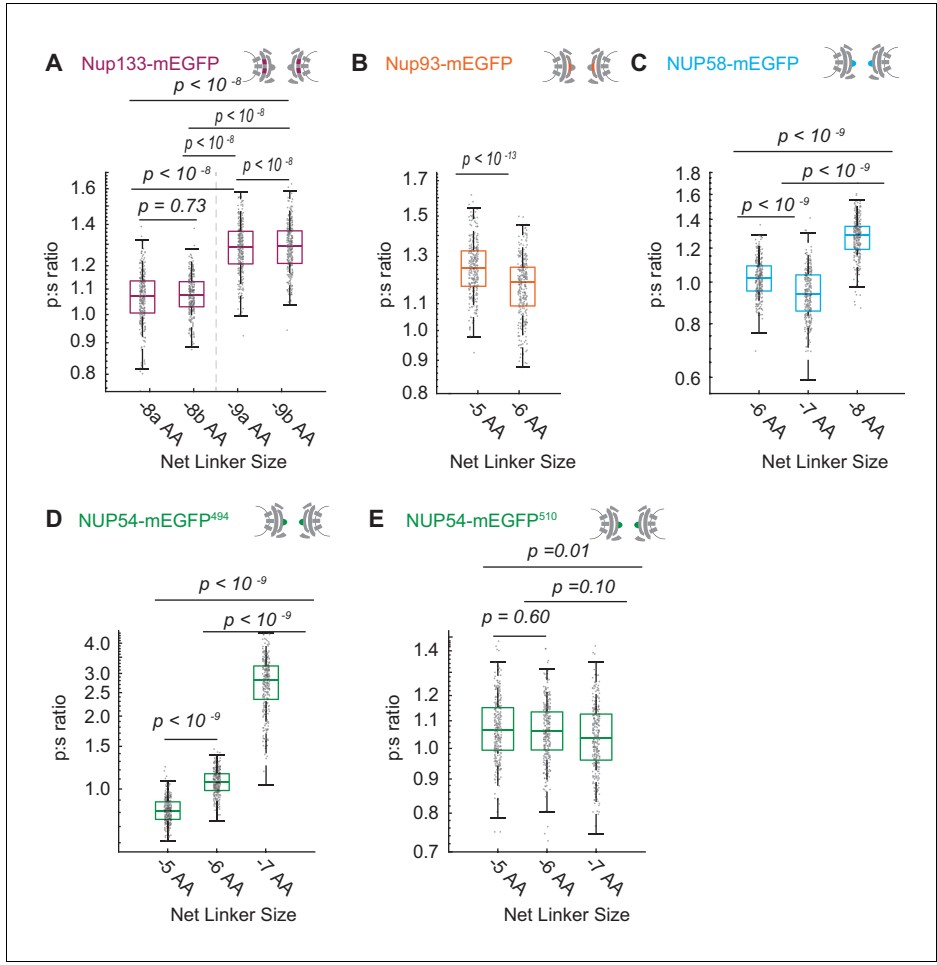

**Figure 2.** Varying the length of the linker between the Nup and the mEGFP by single amino acids to test validity of the orientational sensors. (**A**) Nup133-mEGFP, with linkers of different lengths at its carboxyl terminus, conjugated to mEGFP. A change in the linker length by a single amino acid changes the p:s ratio. Pairs of different rigid alpha-helical linkers of the same length generate indistinguishable p:s ratios. (**B**) Nup93-mEGFP with different linker lengths to mEGFP at the carboxyl terminus. One deleted amino acid shifts the p:s ratio. (**C**) Nup58 with the mEGFP at the carboxyl terminus of the coiled-coiled domain. Each subsequent deletion of a single amino acid alters the p:s ratio. (**D**) Nup54-mEGFP[494] with the mEGFP at the carboxyl terminus of the coiled-coiled domain (Amino Acid: 494). Each subsequent deletion of a single amino acid changes the p:s ratio. (**E**) Nup54-mEGFP[510] with the mEGFP at the carboxyl-terminus of the protein (Amino Acid: 510). Each subsequent deletion of a single amino acid does not alter the p:s ratio. Detailed linker descriptions are available in *Supplementary file 1*. (n = 300 NPCs, 10 cells, boxes indicate quartiles, center bars indicate medians, one-way ANOVA with post-hoc Tukey test for A, C-E, Student's t-test for B).

The online version of this article includes the following source data for figure 2:

**Source data 1.** Source data for *Figure 1*.

change but the orientation of the Nup133 did not shift after starvation. Just as in the transient transfections, Nup54 was reorganized when transport was reduced.

## The orientation of inner ring Nup54 changes after blocking nuclear export

To further test whether attenuating nuclear-cytoplasmic trafficking changes the orientation of domains of Nup54, we used three additional approaches: permeabilization of the plasma membrane with digitonin, treatment with leptomycin B, and expression of dominant-negative Ran.

At a certain concentration, digitonin can selectively permeabilize the plasma membrane while leaving the nuclear membrane intact, resulting in the loss of many cytoplasmic components and the cessation of nuclear-cytoplasmic trafficking (*Adam et al., 1992*). After digitonin permeabilization, we

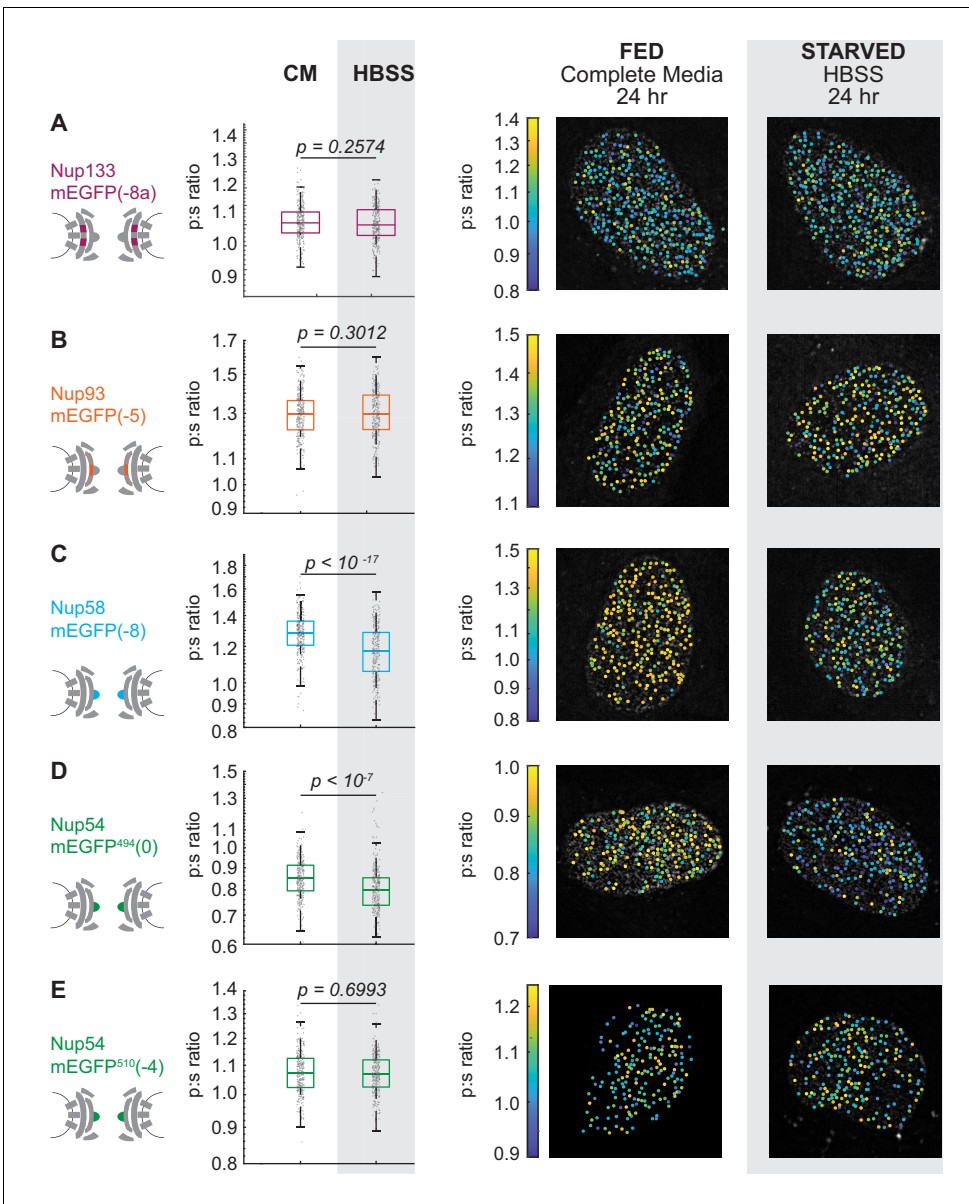

**Figure 3.** The Inner Ring Nups, Nup54 and Nup58, are reorganized with respect to the NPC after starvation. Cells were maintained in complete media (CM) or starved for 24 hr in HBSS prior to imaging. The p:s ratios and representative images are presented for: (**A**) Nup133-mEGFP(−8a), (**B**) Nup93-mEGFP(− 5), (**C**) Nup58-mEGFP(−8), (**D**) Nup54-mEGFP$^{494}$(0), and (**E**) Nup54-mEGFP$^{510}$(−4). The orientation changed for Nup58-mEGFP and Nup54-mEGFP$^{494}$. For additional linker lengths, see *Figure 3—figure supplement 2*. HeLa cells were imaged 48 hr post transfection. (n = 300 NPCs, 10 cells, boxes indicate quartiles, center bars indicate medians, Student's t-test). The online version of this article includes the following source data and figure supplement(s) for figure 3:

**Source data 1.** Source data for *Figure 3*.
**Figure supplement 1.** Nuclear transport is attenuated post-starvation.
**Figure supplement 1—source data 1.** Source data for *Figure 3—figure supplement 1*.
**Figure supplement 2.** The Inner Ring Nups, Nup54 and Nup58, are reorganized with respect to the NPC after starvation.
**Figure supplement 2—source data 1.** Source data for *Figure 3—figure supplement 2*.
**Figure supplement 3.** Cells were maintained in complete media (CM) or starved for 24 hr in HBSS prior to imaging.
**Figure supplement 3—source data 1.** Source data for *Figure 3—figure supplement 3*.

monitored cells to ensure that the nuclear envelope remained intact (*Figure 5—figure supplement 1A*) and we confirmed that the cells were capable of translocating specific NLS-tagged cargos (*Figure 5—figure supplement 1D–E*). When we permeabilize cells and incubate in transport buffer plus 1.5% (wt/vol) 360kD polyvinylpyrrolidone to mimic cytosolic conditions, we observed a distinct shift in the orientation of Nup54$^{494}$ compared to the orientation of Nup54$^{494}$ in unpermeabilized cells grown in complete media. We saw no change in the orientation of Nup133 in permeabilized cells compared to Nup133 in unpermeabilized cells grown in complete media (*Figure 4G–H*).

Leptomycin B blocks nuclear export by inhibiting crm1 (exportin-1), an evolutionarily conserved export factor (*Kudo et al., 1999*). We treated our cells with 25 nM leptomycin B for 15 hr and observed an orientational shift of Nup54$^{494}$ but no change to the orientation of Nup133 (*Figure 4I–J*).

To disrupt both import and export, we expressed a dominant-negative Ran. Ran is a GTPase that mediates nuclear transport (*Moore and Blobel, 1993*). Ran undergoes GTP- hydrolysis in the cytosol that causes the Ran-Kap1-β export complex to dissociate from the NPCs, thereby replenishing Kap1-β in the cytosol. Ran-Q69L is a dominant-negative mutant that cannot perform GTP-hydrolysis, thereby blocking Ran-dependent import and export (*Bischoff et al., 1994*).

We confirmed that cells transiently expressing BFP-RanQ69L for 24 hr were not capable of nuclear-cytoplasmic trafficking by using a light induced nuclear shuttle (*Figure 4M–N*). Nup54$^{494}$ was rearranged once again with respect to the NPC but the Nup133 did not change orientation after cells were transfected with a dominant-negative Ran (*Figure 4K–L*).

These results suggest that attenuating the transport of NLS-driven cargo shifts the orientation of select alpha-helical domains of inner ring Nups. The direction of the p:s ratio shift of Nup54-mEGFP$^{494}$ is consistent among all mechanisms of reducing cargo flux.

## Altering karyopherin content at the NPC changes the conformation of Nup54

After digitonin permeabilization and removal of cytosol, a pool of endogenous karyopherins (kaps) remain associated with the NPCs for a period of hours (*Kapinos et al., 2017*). These kaps can be dissociated from the NPC by introducing Ran-GTP to the nuclear periphery (*Figure 5A*, *Figure 5—figure supplement 1F–H*). Then, the population of kaps can be restored by adding exogenous proteins, which we have purified and have determined to be capable of transporting NLS-tagged cargo (*Figure 5—figure supplement 1D–E*). We have chosen to reintroduce Kap1-β and Kap1-α. The Kap-β (or importin-β) family of kaps encompasses over 20 proteins in mammalian cells. We reintroduced Kap1-β, a 97 kDa import receptor that regulates the canonical import pathway (*Kimura et al., 2017*). Kap1-β interacts with Kap1-α, a 58 kDa adaptor that acts to import classic NLS-tagged cargos (*Pumroy and Cingolani, 2015*).

Removing endogenous kaps shifted the orientation of Nup54$^{494}$ in one direction and restoration of kaps reversed this shift (*Figure 5F–I*). Upon addition of kaps alone, the orientation of Nup54$^{494}$ shifted back towards its position prior to permeabilization. Although the orientation does not totally shift back towards the initial value, we are only supplying two of the many species of kaps present in the NPC in a living cell and no NLS-tagged cargo. In contrast, dissociation or restoration of kaps at the NPC did not produce changes in the orientation of Nup133 (*Figure 5B–E*).

## The arrangement of inner ring Nup62 changes after starvation

To test whether the orientational shifts in Nup54$^{494}$ were coincident with a spatial reorganization of inner ring Nups, we measured the proximity of multiple copies of Nup62 in a single NPC to each other by Förster resonance energy transfer (FRET). We engineered a homozygous cell line where Nup62 was replaced with a copy of Nup62 with FRET sensors (mEGFP$^{290}$ and mCherry$^{321}$) on opposite sides of an alpha-helical domain of Nup62 (*Figure 6*). This alpha helix lies between the coiled-coiled anchor region and a flexible FG-repeat region of Nup62 and is an elongated region that has been predicted to interact with a structured region of Nup54 (*Sharma et al., 2015*). We measured FRET with acceptor photobleaching and quantified FRET efficiency. The FRET efficiency of these sensors was increased post-starvation, consistent with inner ring Nups experiencing a conformational shift (*Figure 6*). The increase in FRET is consistent with the FG-Nup domains being in a more coaligned, physically closer positions and the NPC being in a more constricted state.

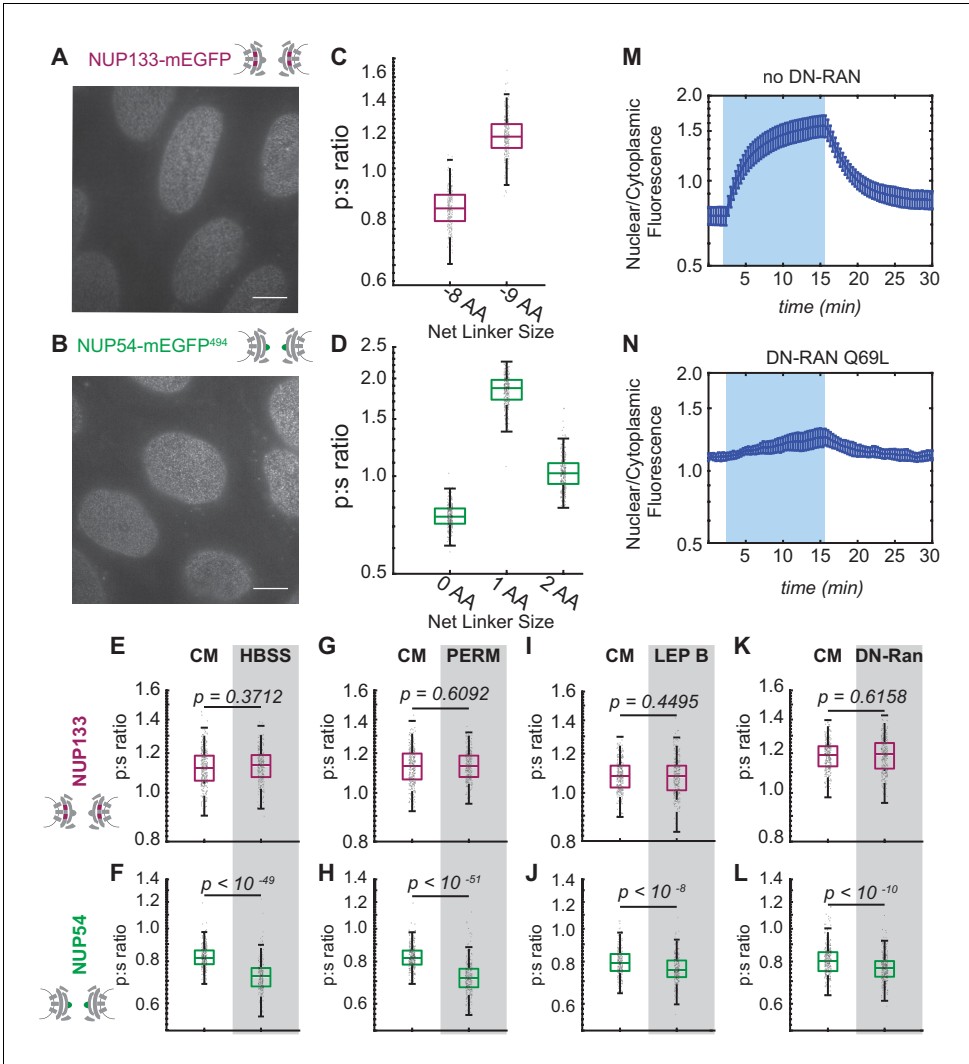

**Figure 4.** Conformational changes of the Inner Ring of the NPC revealed by perturbations of cargo state in CRISPR cell lines. (**A-B**) No morphological distortions are detected in cell lines endogenously expressing orientational sensors. (**C**) Nup133-mEGFP cell lines with the mEGFP at the carboxyl-terminus of the protein with different linker lengths. One deleted amino acid shifts the p:s ratio. (**D**) Nup54-mEGFP$^{494}$ cell lines with the mEGFP at the carboxyl-end of the coiled-coiled domain of the protein with constructs of different linker lengths. One deleted amino acid shifts the p:s ratio. Detailed linker descriptions are available in *Supplementary file 2*. (**E-F**) Orientational sensor cell lines were maintained in CM or starved for 24 hr in HBSS. (**G-H**) CRISPR cell lines mock permeabilized or digitonin-permeabilized prior to imaging. (**I-J**) CRISPR cell lines mock treated or treated with leptomycin B prior to imaging. (**K-L**) CRISPR cell lines with or without transient expression of dominant-negative Ran. (n = 300 NPCs, 10 cells, boxes indicate quartiles, center bars indicate medians, Student's t-test). (**M-N**) The average change in the ratio of nuclear/cytoplasmic fluorescence in HeLa cells with and without dominant-negative Ran transiently expressed (n = 6, mean reported ± SEM with error bars). The shaded region represents the time of blue light LANS activation.

The online version of this article includes the following source data and figure supplement(s) for figure 4:

**Source data 1.** Source data for *Figure 4C–L*.
**Source data 2.** Source data for *Figure 4M–N*.
**Figure supplement 1.** Validation of CRISPR cell lines.
**Figure supplement 2.** The Inner Ring Nup Nup54 is reorganized with respect to the NPC after starvation.
**Figure supplement 2—source data 1.** Source data for *Figure 4—figure supplement 2*.

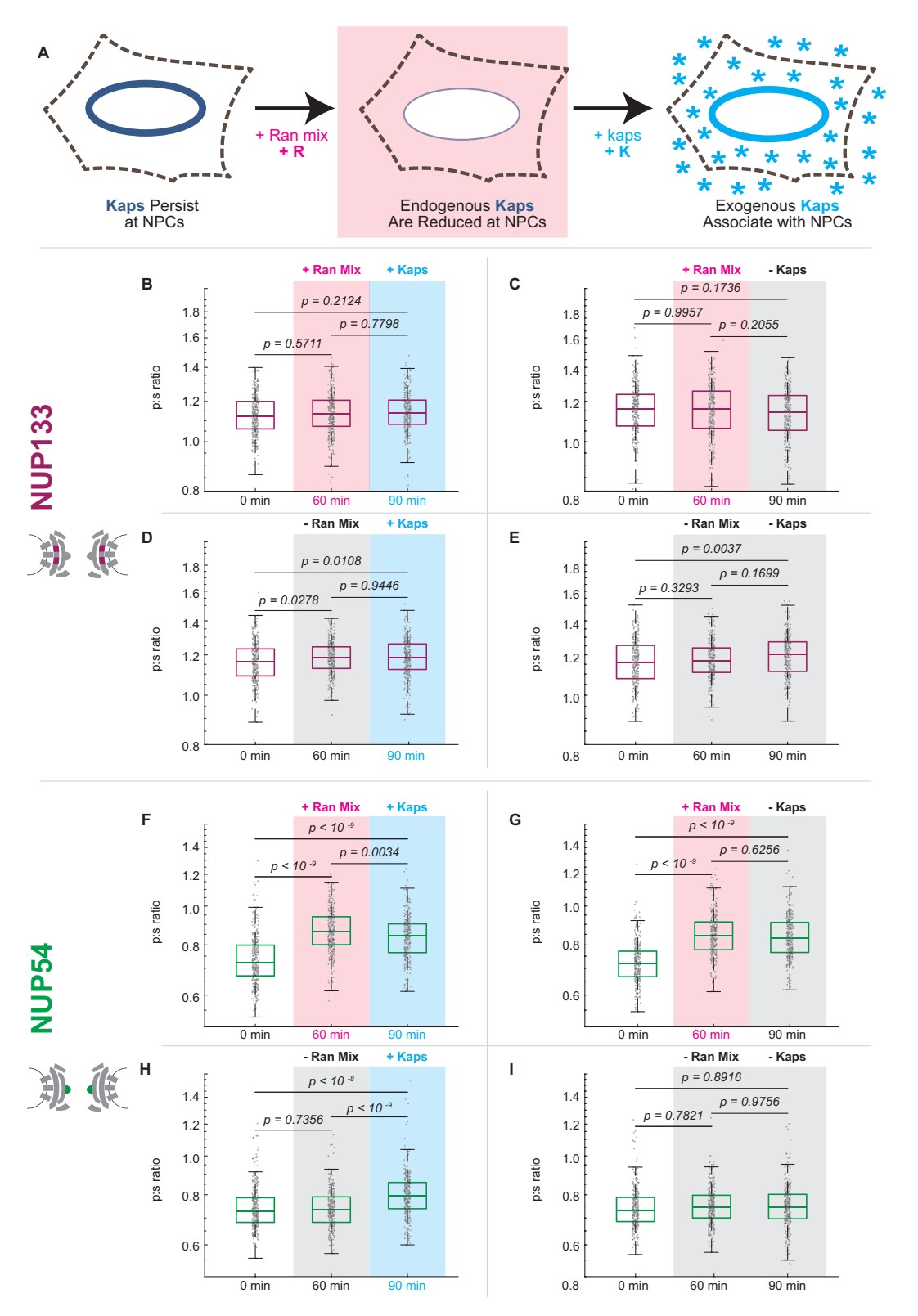

**Figure 5.** Karyopherin content at the nuclear periphery induces conformational changes in Nup54-mEGFP[494] but not Nup133-mEGFP. (**A**) Digitonin permeabilization allows the introduction of transport factors to the nuclear periphery. (**B-E**) Nup133-mEGFP does not experience a shift in orientation after removal of endogenous kaps or addition of exogenous kaps. (**F-I**) The orientation of Nup54-mEGFP[494] changes after removal of endogenous kaps or introduction of exogenous kaps. Pink boxes indicate the addition of Ran mix, blue boxes indicate the addition of kaps, and gray boxes indicate a

*Figure 5 continued on next page*

*Figure 5 continued*

buffer change with no additional transport factors. (n = 300 NPCs, 30 cells, boxes indicate quartiles, center bars indicate medians, one-way ANOVA with post-hoc Tukey test).

The online version of this article includes the following source data and figure supplement(s) for figure 5:

**Source data 1.** Source data for *Figure 5*.
**Figure supplement 1.** Introduction of nuclear transport factors to the nuclear periphery using digitonin permeabilization of the plasma membrane.
**Figure supplement 1—source data 1.** Source data for *Figure 5—figure supplement 1*.

## Discussion

A number of recent studies have reported variability in the organization and diameter of NPCs. Super-resolution imaging has been used to visualize NPCs at different developmental stages of *X. laevis* oocytes, demonstrating that the organization and diameter of the NPCs changes over time (*Sellés et al., 2017*). By cryo-ET using sub-tomogram averaging in either transport competent or transport inhibited cells, two distinct structural states are observed with differences in the central transporter, suggesting that this region might undergo conformational changes upon engagement of cargo (*Eibauer et al., 2015*; *Zwerger et al., 2016*). In HeLa cells, NPCs from the same cell were observed with cryo-ET to be more similar in inner diameter than those from other cells, suggesting that the diameter might change as a result of a cell's specific physiological state (*Mahamid et al., 2016*). While this manuscript was in review, cryo-EM tomograms revealed *in situ* NPCs taken from *S. cerevisiae* cells in exponential growth phase were ~20 nm larger in diameter than isolated NPCs from *S. cerevisiae*, underscoring the potential flexibility of the NPC (*Allegretti et al., 2020*). Structural studies based on two discrete crystal states of short peptides of three inner channel ring Nups, Nup58, Nup54, and Nup62, have led to the proposal that the structured regions of these inner ring Nups cycle between a dilated ring of 40–50 nm in diameter and a constricted ring of 20 nm in diameter. This Ring Cycle hypothesis suggests that this ring undergoes conformational changes and directly regulates cargo import and export (*Melcák et al., 2007*; *Sharma et al., 2015*; *Solmaz et al., 2013*; *Solmaz et al., 2011*). It has also been proposed that the evolution of the NPC into a complex sized over ~109 MDa was in part driven by the need to cushion the huge diameter changes of the central transport channel by a large and deformable surrounding protein matrix (*Hoelz et al., 2011*).

The conformational changes we observe in the NPCs of living cells may be coincident with cargo translocation or may be an indication that the scaffold of the NPC serves as a dynamic gate that can regulate nuclear trafficking. These conformational changes are consistent with observations that NPC diameter is altered in HeLa cells under different physiological conditions (*Mahamid et al., 2016*) and with observations of NPCs with different diameters from *S. cerevisiae* (*Allegretti et al., 2020*). Our results are also consistent with the Ring Cycle hypothesis. Further studies will be needed

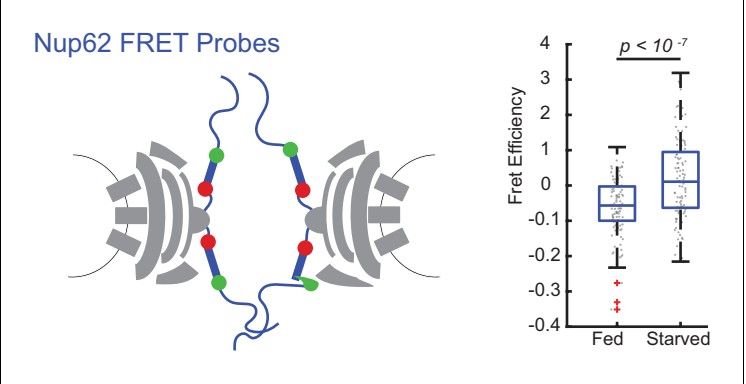

**Figure 6.** FRET between Nup62 'finger' domains increases after starvation. Under starvation conditions, FRET increased between Nup62 'finger' domains. (A) Schematic of Nup62 FRET probe labeling scheme. (B) FRET efficiency for HeLa cells were imaged 48 hr post transfection. Cells were kept in CM or starved for 24 hr in HBSS (n = 100 NPCs, 10 cells, boxes indicate quartiles, bars indicate medians, Student's t-test).

The online version of this article includes the following source data for figure 6:

**Source data 1.** Source data for *Figure 6*.

to determine the exact conformations we are monitoring in this study, and to what extent the conformations we observe align with these models. In addition to changes in orientation, we also observe increases in FRET in the FG-regions of Nup62 upon starvation, which is consistent with a spatial rearrangement and constriction of the NPC diameter. The NPC structure as a whole may be modulated by cargo load and identity. Compositional and structural flexibilities in the NPC underscore the complexity of this macromolecular complex, and their effects on transport and regulation remain exciting future avenues of research in the field.

The experiments we present have been done at the level of single active NPCs in which altered conformations are correlated to the loading of kaps. Our results are consistent with orientational changes to the inner ring of the NPC in response to changes in transport state. Although we cannot yet calculate the magnitude of the angle changes of these alpha helices, we can say with confidence that these alpha helices are experiencing a rearrangement with respect to the NPC after they are confronted with perturbations in transport. Further, these orientational shifts are confined to the inner ring, the most constricted portion of the NPC: similar changes are not detected when probes are placed in other locations. It is possible that those regions are moving translationally and thus escaping our detection, which is sensitive only to orientation changes with respect to the nucleo-cytoplasmic axis.

The imaging technique presented here allows for the monitoring of Nup orientational changes within many individual NPCs simultaneously in a single living cell. We used this approach to look at dynamics of assembly of HIV-1 *in vivo* (*Johnson et al., 2018*). This technique builds upon on our previous work (*Atkinson et al., 2013*; *Kampmann et al., 2011*; *Mattheyses et al., 2010*) to provide tools to directly observe the conformations and orientational dynamics of Nups in living cells. By expanding this technique into pol-TIRFM, we have developed tools to that allow us to directly observe the orientational changes of Nups within a single NPC in living cells and to track these changes over time with different cargo conditions.

This light microscopy technique can be used to probe the orientations of different domains within an individual protein either on its own or as part of a macromolecular complex. By allowing one to not only localize a protein, but to also monitor the orientations of different domains within a protein, this technique can provide insights into various molecular mechanisms. Polarization microscopy can be used to monitor the dynamics and organization of domains of other macromolecular complexes *in vivo* and *in vitro*, including the ribosome, proteasome, centriole, and cilia. Although to use this technique *in vivo*, the macromolecular complex to be studied would need to have an axis that can be defined with respect to the illumination field, these geometric limitations only exist *in vivo*. Using nanofabrication techniques, a macromolecular complex or substrate can be conjugated with respect to the coverslip and the orientation of a dynamic domain can be monitored. Although the technique currently requires a carboxyl-terminal alpha helix of a peptide domain of interest to be conjugated to the amino alpha helix of a fluorescent protein, development of orientationally confined fluorescent probes will allow the domain of any protein to be tagged and monitored.

# Materials and methods

**Key resources table**

| Reagent type (species) or resource | Designation | Source or reference | Identifiers | Additional information |
|---|---|---|---|---|
| Antibody | α- karyopherin α1/6 (2D9) (rat monoclonal) | Santa Cruz | sc-101540 RRID:AB_2133549 | IF 1:500 |
| Antibody | α-kap1ß/imp ß −1 (3E9) (mouse monoclonal) | Abcam | ab2811 RRID:AB_2133989 | IF 1:1000 |

*Continued on next page*

*Continued*

| Reagent type (species) or resource | Designation | Source or reference | Identifiers | Additional information |
|---|---|---|---|---|
| Antibody | anti-Rabbit IgG (H+L) Cross-Adsorbed Secondary Antibody, Alexa Fluor 488 (goat polyclonal) | Invitrogen | CAT # A-11008 RRID:AB_2534074 | IF 1:2000 |
| Antibody | anti-Mouse IgG (H+L) Cross-Adsorbed Secondary Antibody, Alexa Fluor 594 (goat polyclonal) | Invitrogen | CAT # A-11005 RRID:AB_141372 | IF 1:2000 |
| Antibody | anti-GFP: Living Colors A.v. Monoclonal Antibody (JL-8) | Clontech | CAT # 632381 RRID:AB_2313808 | WB 1:1000 |
| Antibody | anti-Mouse IgG (Fab specific)– Peroxidase antibody (goat polyclonal) | Sigma-Aldrich | CAT # A9917 RRID:AB_258476 | WB 1:50,000 |
| Antibody | Anti-β-Actin antibody, Ac-74 (mouse monoclonal) | Sigma-Aldrich | CAT # A5316 RRID:AB_476743 | WB 1:1000 |
| Strain, strain background (*Escherichia coli*) | BL21-CodonPlus (DE3)-RIL | Agilent | CAT # 230245 | Chemically competent cells |
| Chemical compound, drug | Dulbecco's Modified Eagle's Medium | Gibco | CAT # 11995–065 | |
| Chemical compound, drug | Fetal Bovine Serum | Sigma-Aldrich | CAT #F4135 | |
| Chemical compound, drug | Hanks Balanced Salt Solution with Calcium and Magnesium | Gibco | CAT # 14025076 | |
| Chemical compound, drug | Leptomycin B | Sigma-Aldrich | CAT # L2913 | 25 nM |
| Chemical compound, drug | Fibronectin | Gibco | CAT # 33010018 | |
| Chemical compound, drug | Paraformaldehyde | Electron Microscopy Sciences | CAT #15711 | 4% w/v |
| Chemical compound, drug | PBS, pH 7.4 | Gibco | CAT # 10010023 | |
| Chemical compound, drug | Normal Donkey Serum | Sigma-Aldrich | CAT # 566460 | 2.5% v/v |
| Chemical compound, drug | Normal Goat Serum | Sigma-Aldrich | CAT # NS02L | 2.5% v/v |
| Chemical compound, drug | Bovine Serum Albumin | Sigma-Aldrich | CAT #A2153 | 1% v/v |
| Chemical compound, drug | SlowFade Diamond Antifade Mountant | Invitrogen | CAT # S36972 | |
| Chemical compound, drug | FuGENE 6 Transfection Reagent | Promega | CAT #E2691 | |
| Chemical compound, drug | Opti-MEM I Reduced Serum Medium, no phenol red | Gibco | CAT # 11058021 | |
| Chemical compound, drug | Digitonin, High Purity – Calbiochem | Millipore | CAT # 300410; CAS 11024-24-1 | 34 µg/mL |

*Continued on next page*

*Continued*

| Reagent type (species) or resource | Designation | Source or reference | Identifiers | Additional information |
|---|---|---|---|---|
| Chemical compound, drug | 360kD polyvinylpyrrolidone (PVP) | Sigma-Aldrich | CAT #PVP360; CAS 9003-39-8 | 1.5% w/v |
| Chemical compound, drug | R-phycoerythrin | ThermoFisher | CAT # P801 | 500 ng/mL |
| Chemical compound, drug | isopropyl β-D-1-thiogalactopyranoside (IPTG) | Sigma-Aldrich | CAT # I5502 | 0.5 mM |
| Chemical compound, drug | Benzonase Nuclease | EMD Millipore | CAT # 70746 | 25 U/mL |
| Chemical compound, drug | rLysozyme Solution | EMD Millipore | CAT # 71110 | 12 U/mL |
| Chemical compound, drug | cOmplete, EDTA-free Protease Inhibitor Cocktail | Roche | CAT # 11873580001 | |
| Chemical compound, drug | Imidazole | Alfa Aesar | CAT #47274; CAS 288-32-4 | |
| Chemical compound, drug | Ni-NTA Agarose | Qiagen | CAT # 30250 | |
| Chemical compound, drug | Guanosine-5′-Triphosphate Disodium Salt | Fisher Scientific | CAT # AAJ16800MC; CAS 56001-37-7 | 0.1 mM |
| Chemical compound, drug | Adenosine 5′-triphosphate disodium salt (ATP disodium salt) hydrate | VWR | CAT # TCA0157; CAS 34369-07-8 | 1 mM |
| Chemical compound, drug | Creatine phosphate | Sigma-Aldrich | CAT # CRPHO-RO; CAS 71519-72-7 | 1 mg/mL |
| Chemical compound, drug | Creatine Phosphokinase, Porcine Heart | Sigma-Aldrich | CAT # 238395; CAS 9001-15-4 | 15 U/mL |
| Chemical compound, drug | NucBlue Live ReadyProbes Reagent (Hoechst 33342) | ThermoFisher | CAT # R37605 | |
| Chemical compound, drug | FuGENE HD Transfection Reagent | Promega | CAT # E2311 | |
| Chemical compound, drug | Puromycin | Invivogen | CAT # ant-pr-5 | |
| Cell line (*Homo-sapiens*) | HeLa Cells | ATCC | CCL-2 RRID:CVCL_0030 | |
| Cell line (*Homo-sapiens*) | Hap1 Cells | Horizon | N/A | |
| Cell line (*Homo-sapiens*) | Nup133_mEGFP(−9) | This Paper | N/A | CRISPR-edited Hap1 cell line expressing Nup133_mEGFP(−9) |
| Cell line (*Homo-sapiens*) | Nup133_mEGFP(−8) | This Paper | N/A | CRISPR-edited Hap1 cell line expressing Nup133_mEGFP(−8) |
| Cell line (*Homo-sapiens*) | Nup54-mEGFP$^{494}$(0) | This Paper | N/A | CRISPR-edited Hap1 cell line expressing Nup54-mEGFP$^{494}$(0) |

*Continued on next page*

*Continued*

| Reagent type (species) or resource | Designation | Source or reference | Identifiers | Additional information |
|---|---|---|---|---|
| Cell line (*Homo-sapiens*) | Nup54-mEGFP[494](1) | This Paper | N/A | CRISPR-edited Hap1 cell line expressing Nup54-mEGFP[494](1) |
| Cell line (*Homo-sapiens*) | Nup54-mEGFP[494](2) | This Paper | N/A | CRISPR-edited Hap1 cell line expressing Nup54-mEGFP[494](2) |
| Cell line (*Homo-sapiens*) | Nup62_mCherry290_mEGFP321 | This Paper | N/A | CRISPR-edited Hap1 cell line expressing Nup62_mCherry 290_mEGFP321 |
| Sequence-based reagent | Cloning Primer Nup54 CRISPR Forward | This Paper | PCR primers | CACC**CGATCTAGA AGATATAAAGC** (guide bolded) |
| Sequence-based reagent | Cloning Primer Nup54 CRISPR Reverse | This Paper | PCR primers | AAACGCTTTATAT CTTCTAGATCG |
| Sequence-based reagent | Cloning Primer Nup133 CRISPR Forward | This Paper | PCR primers | CACC**GCTCAGTGA GTACTTACCGG** (guide bolded) |
| Sequence-based reagent | Cloning Primer Nup133 CRISPR Reverse | This Paper | PCR primers | AAACCCGGTAAGT ACTCACTGAGC |
| Sequence-based reagent | PCR Primer Nup54 Forward | This Paper | PCR primers | CCTGTGACTA GCTTGCAGTT |
| Sequence-based reagent | PCR Primer Nup54 Reverse | This Paper | PCR primers | ACCTCTGATGT GGATGGTTTC |
| Sequence-based reagent | PCR Primer Nup133 Forward | This Paper | PCR primers | AGTCCAATCCTT ACTTCGAGTTT |
| Sequence-based reagent | PCR Primer Nup133 Reverse | This Paper | PCR primers | AGGAACAACAAC TGACACATTTC |
| Recombinant DNA reagent | Nup133_mEGFP(−8a) (plasmid) | *Kampmann et al., 2011* | Addgene # 163417 | Mammalian expression of Nup133 fused at carboxy-terminus to mEGFP with total net fusion of (−8 amino acids) |
| Recombinant DNA reagent | Nup133_mEGFP(−8b) (plasmid) | *Kampmann et al., 2011* | Addgene #163418 | Mammalian expression of Nup133 fused at carboxy-terminus to mEGFP with total net fusion of (−8 amino acids) |
| Recombinant DNA reagent | Nup133_mEGFP(−9a) (plasmid) | *Kampmann et al., 2011* | Addgene # 163419 | Mammalian expression of Nup133 fused at carboxy-terminus to mEGFP with total net fusion of (−9 amino acids) |
| Recombinant DNA reagent | Nup133_mEGFP(−9b) (plasmid) | *Kampmann et al., 2011* | Addgene # 163420 | Mammalian expression of Nup133 fused at carboxy-terminus to mEGFP with total net fusion of (−9 amino acids) |
| Recombinant DNA reagent | Nup93_mEGFP(−5) (plasmid) | This paper | Addgene # 163421 | Mammalian expression of Nup93 fused at carboxy-terminus to mEGFP with total net fusion of (−5 amino acids) |

*Continued on next page*

*Continued*

| Reagent type (species) or resource | Designation | Source or reference | Identifiers | Additional information |
|---|---|---|---|---|
| Recombinant DNA reagent | Nup93_mEGFP(−6) (plasmid) | This paper | Addgene # 163422 | Mammalian expression of Nup93 fused at carboxy-terminus to mEGFP with total net fusion of (−6 amino acids) |
| Recombinant DNA reagent | Nup58_mEGFP(−6) (plasmid) | This paper | Addgene # 163423 | Mammalian expression of Nup58 with mEGFP (missing first six amino acids) at position 412 |
| Recombinant DNA reagent | Nup58_mEGFP(−7) (plasmid) | This paper | Addgene # 163424 | Mammalian expression of Nup58 with mEGFP (missing first seven amino acids) at position 412 |
| Recombinant DNA reagent | Nup58_mEGFP(−8) (plasmid) | This paper | Addgene # 163425 | Mammalian expression of Nup58 with mEGFP (missing first eight amino acids) at position 412 |
| Recombinant DNA reagent | Nup54-mEGFP$^{494}$(0) (plasmid) | This paper | Addgene # 163426 | Mammalian expression of Nup54 with mEGFP (missing first five amino acids) at amino acid 494 with five amino acid rigid alpha helical linker |
| Recombinant DNA reagent | Nup54-mEGFP$^{494}$(1) (plasmid) | This paper | Addgene # 163427 | Mammalian expression of Nup54 with mEGFP (missing first five amino acids) at amino acid 494 with six amino acid rigid alpha-helical linker |
| Recombinant DNA reagent | Nup54-mEGFP$^{494}$(2) (plasmid) | This paper | Addgene # 163428 | Mammalian expression of Nup54 with mEGFP (missing first five amino acids) at amino acid 494 with seven amino acid rigid alpha helical linker |
| Recombinant DNA reagent | Nup54-mEGFP494(flex0) (plasmid) | This paper | Addgene # 163429 | Mammalian expression of Nup54 with mEGFP (missing first five amino acids) at amino acid 494 with five amino acid flexible alpha-helical linker |
| Recombinant DNA reagent | Nup54-mEGFP494(flex1) (plasmid) | This paper | Addgene # 163430 | Mammalian expression of Nup54 with mEGFP (missing first five amino acids) at amino acid 494 with six amino acid flexible alpha-helical linker |
| Recombinant DNA reagent | Nup54-mEGFP494(flex2) (plasmid) | This paper | Addgene # 163431 | Mammalian expression of Nup54 with mEGFP (missing first five amino acids) at amino acid 494 with seven amino acid flexible alpha helical linker |
| Recombinant DNA reagent | Nup54_mEGFP$^{494}$(−4) (plasmid) | This paper | Addgene # 163432 | Mammalian expression of Nup54 with mEGFP (missing first four amino acids) at amino acid 494 |
| Recombinant DNA reagent | Nup54_mEGFP$^{494}$(−5) (plasmid) | This paper | Addgene # 163433 | Mammalian expression of Nup54 with mEGFP (missing first five amino acids) at amino acid 494 |

*Continued on next page*

Continued

| Reagent type (species) or resource | Designation | Source or reference | Identifiers | Additional information |
|---|---|---|---|---|
| Recombinant DNA reagent | Nup54_mEGFP$^{494}$(−6) (plasmid) | This paper | Addgene # 163434 | Mammalian expression of Nup54 with mEGFP (missing first six amino acids) at amino acid 494 |
| Recombinant DNA reagent | Nup54_mEGFP$^{510}$(−4) (plasmid) | This paper | Addgene # 163435 | Mammalian expression of Nup54 with mEGFP (missing first five amino acids) at the carboxy-terminus with total net fusion of (−4) amino acids |
| Recombinant DNA reagent | Nup54_mEGFP$^{510}$(−5) (plasmid) | This paper | Addgene # 163436 | Mammalian expression of Nup54 with mEGFP (missing first five amino acids) at the carboxy-terminus with total net fusion of (−5) amino acids |
| Recombinant DNA reagent | Nup54_mEGFP$^{510}$(−6) (plasmid) | This paper | Addgene # 163437 | Mammalian expression of Nup54 with mEGFP (missing first five amino acids) at the carboxy-terminus with total net fusion of (−6) amino acids |
| Recombinant DNA reagent | pSpCas9(BB)−2A-Puro (PX459) V2.0 (plasmid) | *Ran et al., 2013* | Addgene # 62988 | |
| Recombinant DNA reagent | pTriEx-mCherry::LANS4 (plasmid) | *Yumerefendi et al., 2015* | Addgene #60785 | |
| Recombinant DNA reagent | BFP-RanQ69L (plasmid) | This paper | Addgene # 163438 | Mammalian expression of RanQ69L with tag-BFP |
| Recombinant DNA reagent | pET28-RAN (plasmid) | Günter Blobel | Addgene # 163439 | Ran in pET28 protein expression backbone |
| Recombinant DNA reagent | pET28_KPNA1 (plasmid) | This paper | Addgene #163440 | KPNA1 in pET28 protein expression backbone |
| Recombinant DNA reagent | pET28-KPNB1 (plasmid) | This paper | Addgene # 163441 | KPNB1 in pET28 protein expression backbone |
| Recombinant DNA reagent | pET28-NTF2 (plasmid) | This paper | Addgene # 163442 | NTF2 in pET28 protein expression backbone |
| Recombinant DNA reagent | Nup54 no FG-mEGFP494(0) | This paper | Addgene # 164269 | Mammalian expression of Nup54 without the FG-Nup domain, with mEGFP (missing the first 5 amino acids) at amino acid position 494 with a rigid alpha helix of 5 amino acids at the carboxyl end of mEGFP |
| Recombinant DNA reagent | Nup54 no FG-mEGFP494(1) | This paper | Addgene # 164270 | Mammalian expression of Nup54 without the FG-Nup domain, with mEGFP (missing the first 5 amino acids) at amino acid position 494 with a rigid alpha helix of 6 amino acids at the carboxyl end of mEGFP |
| Recombinant DNA reagent | Nup54 no FG-mEGFP494(2) | This paper | Addgene # 164271 | Mammalian expression of Nup54 without the FG-Nup domain, with mEGFP (missing the first 5 amino acids) at amino acid position 494 with a rigid alpha helix of 7 amino acids at the carboxyl end of mEGFP |

*Continued on next page*

*Continued*

| Reagent type (species) or resource | Designation | Source or reference | Identifiers | Additional information |
|---|---|---|---|---|
| Recombinant DNA reagent | NLS-tdTomato | This paper | Addgene # 163443 | Bacterial expression of His-tagged, SV40 NLS-tagged tdTomato in the modified pRSETB protein expression backbone |
| Software, algorithm | Metamorph Ver 7.7.8 | Molecular Devices | https://www.molecular devices.com/products/ cellular-imaging-systems/acquisition -and-analysis-software/metamorph- microscopy#gref | |
| Software, algorithm | MatLab 2019A | Mathworks | https://www. mathworks.com/ | |
| Software, algorithm | Fiji | *Schindelin et al., 2012* | https://imagej.net/Fiji/ Downloads | |
| Software, algorithm | CRISPR Guide RNA Design | Benchling | https://www. benchling.com/crispr/ | |
| Software, algorithm | Adobe Illustrator | Adobe | https://www.adobe.com/ products/illustrator.html | |

## Cell lines and growth

HeLa (ATCC, CCL-2), Hap1 (Horizon Discovery, Cambridge, UK), and Hap1-derived Nup-mEGFP cell lines (this paper) were cultured in Dulbecco's Modified Eagle's Medium (DMEM, Gibco, Waltham, MA), supplemented with l-glutamine and sodium pyruvate (from here-on referred to as DMEM) and 10% (vol/vol) fetal bovine serum (FBS, Sigma, St. Louis, MO) in humidified incubators at 37C and in a 5% $pCO_2$ atmosphere, using standard sterile techniques. HeLa cells were recently acquired from ATCC and Hap1 cell lines were acquired from Horizon Discovery. Cells were negative for mycoplasma.

For starvation experiments, cells were imaged, then washed 3x with PBS and placed in 1x Hank's Balanced Salt Solution with calcium and magnesium (HBSS, Gibco) for 24 hr and imaged. For leptomycin B experiments, cells were either treated with 25 nM leptomycin B (Sigma) or vector (methanol) for 15 hr, at which point both were imaged.

## Imaging conditions

Cells were seeded onto MatTek dishes with no. 1.5 coverslips. For HeLa cells, the dishes were uncoated, but for Hap1 cells (all CRISPR cell lines) the dishes were coated with fibronectin (Gibco). During imaging, the media was replaced with cell imaging media [HBSS (Sigma), 10 mM HEPES, pH7.4], supplemented with 10% FBS (vol/vol, Sigma).

## Microscopy: general

Cells were imaged on a custom-built microscope, based on an inverted IX-81 frame (Olympus Life Sciences, Tokyo, Japan) and equipped with a custom-built through-the-objective (100X UAPON 1.49 NA, Olympus and 100x UAPON 1.51 NA, Olympus Life Sciences) polarized TIRFM illuminator equipped with a 405 nm laser (100 mW LuxX diode laser, Omicron, Rodgau-Dudenhofen, Germany), a 488 nm laser (100 mW LuxX diode laser, Omicron), a 594 nm laser (100 mW diode-pumped solid-state laser, Cobolt AB, Stockholm, Sweden), and a 647 nm laser (100 mW LuxX diode laser, Omicron) (*Johnson et al., 2014*). For live-cell and permeabilized-cell experiments, the temperature was maintained at 37C throughout imaging using custom-built housing.

The excitation TIR light was azimuthally scanned at 200 Hz with mirror galvanometers (Nutfield Technology, Cranberry Township, PA). An electro-optic modifier (EOM, Conoptics, Danbury, CT) and a quarter-wave-plate (Thorlabs, Newton, NJ) before the galvanometers controlled the polarization of the 488 nm laser.

The galvanometers, EOM, camera shutter, and 488 laser shutter were all driven by a multifunctional data acquisition board (PCIe-6323, 577, National Instruments, Austin, TX) and controlled from

custom written software in LabView (National Instruments) (*Johnson et al., 2014*). All emission light was collected after it was passed through a multiband polychroic (zt405/488/594/647rpc 2 mm substrate, Chroma, Bellows Falls, VT) to isolate the excitation light from the emitted light.

Images were collected on a CMOS camera (Flash-4.0, Hamamatsu Photonics, Middlesex, NJ) connected with Hamamatsu Camera Link interface to a workstation (Precision Model T7500, Dell, Austin, TX) running image acquisition software (Metamorph, Molecular Devices, San Jose, CA) (*Johnson et al., 2014*).

## Microscopy: pol-TIRFM of mEGFP Nups

A sequence of 20 images was taken with alternating $\hat{p}$- and $\hat{s}$-excitation in TIR. Each individual $\hat{p}$ or image had an exposure time of 5 ms (laser power: 100 mW), and a new image was collected every 15 ms.

All image analysis was automated with author-written analysis algorithms written in MATLAB. 10 $\hat{p}$ and 10 $\hat{s}$ images for a given timepoint were summed to form a single $\hat{p}$ and a single $\hat{s}$ image for each time point. Camera background was subtracted from each image. A region of the cell containing the nucleus was chosen by the user for automated detection and identification of NPCs, to prevent the analysis algorithm from considering any cytoplasmic puncta. NPCs were identified via an automated algorithm (*Pulupa, 2020*) using the Laplacian of Gaussian (LoG) algorithm written by *Garcia, 2020*, based on *Lindeberg, 1998*. NPCs were excised from a background subtracted image (top-hat filtered), and both polarizations were fit to a Gaussian. If either polarization did not fit to a Gaussian, the data point was rejected. The intensities from the original (not-top-hat filtered but camera background subtracted and summed) images were then extracted for analysis by taking the value from the maximum intensity pixel from each punctum at each polarization. A p:s ratio was then calculated for each punctum.

## Microscopy: immunofluorescence

Cells were imaged with 488 laser (laser power: 5 mW) and 594 laser (laser power: 5 mW) for 200 ms.

Cells were grown on MatTek dishes at 37C in 5% $_{p}CO_2$. Cells were fixed with 4% (wt/vol) paraformaldehyde in PBS for 15 min at room temperature. They were then washed 3 times for 5 min in PBS, and then permeabilized with 0.1% (vol/vol) Triton X-100 in PBS for 10 min. Cells were then blocked for 1 hr in blocking buffer (0.1% vol/vol) Triton X-100, 2.5% normal donkey serum, 2.5% normal goat serum, and 1% BSA (all from Sigma). Primary antibodies were then added in blocking buffer and incubated overnight at 4C in a humid chamber. Cells were then washed 3 times for 5 min in PBS and then incubated with secondary antibody in 0.1% (vol/vol) Triton X-100 in PBS for 1–2 hr at room temperature. Dishes were dried and mounted with SlowFade Diamond Antifade Mountant (Invitrogen, Waltham, MA). Antibodies used: monoclonal rat-α-karyopherin α1/6 (2D9, Santa Cruz, Santa Cruz, CA, RRID:AB_2133549) at 1:500, monoclonal mouse-α-kap1β/impβ−1 (3E9, Abcam, Cambridge, MA, RRID:AB_2133989) at 1:1000, goat α-rat (AF488, RRID:AB_2534074) at 1:2000, goat α-mouse (AF594, RRID:AB_141372) from Invitrogen (1:2000). Images were quantified using FIJI software (National Institutes of Health, Bethesda, MD) (*Schindelin et al., 2012*). The nuclear rim was defined as a region of interest by thresholding the fluorescent image of karyopherin 1β (Kap1β) and converting into a binary image. The image was then used to form a mask by: filling holes, eroding, outlining, and dilating. This mask was then used to quantify intensity from Kap1β and Kap1α channels.

## Microscopy: light activated nuclear shuttle (LANS)

Cells were imaged every 20 s with 594 laser (laser power: 5 mW, 200 ms exposure time) for 2 min, then 594 laser acquisitions (laser power: 5 mW, 200 ms exposure time) were interleaved with pulses of 488 acquisitions (laser power: 3 mW, 2 s exposure time) every 20 s for 13 min, and then cells were imaged every 20 s with 594 excitation light (laser power: 5 mW, 200 ms exposure time) for 15 min with no 488 excitations. All excitations were done in a 'semi'-TIRF excitation mode, which restricts fluorescence to a few micrometers near to the coverslip to illuminate more of the cytosol. Regions of the nucleus and the cytosol were manually selected in FIJI (*Schindelin et al., 2012*), avoiding fluorescent aggregates, and quantified for determination of the nuclear fluorescence/cytoplasmic fluorescence over time.

## Microscopy: FRET

Cells were seeded onto fibronectin-coated MatTek dishes as described above. For starvation conditions, cells were grown 1 day then washed 3x with PBS and placed in 1x Hank's Balanced Salt Solution with calcium and magnesium (HBSS, Gibco) for 24 hr and imaged. For control condition, cells were grown 2 days then placed in cell imaging media as described above. Cells were imaged with the 488 laser (laser power: 15 mW, 200 ms exposure time) and the 594 laser (laser power: 30 mW, 200 ms exposure time). The mCherry was then bleached by imaging every second with the 594 laser (laser power: 30 mW, 950 ms exposure time) for 1 min. Cells were then imaged again with the 488 laser (laser power: 15 mW, 200 ms exposure time) and the 594 laser (laser power: 30 mW, 200 ms exposure time).

## FRET quantification

NPCs are manually identified in pre-bleached images in Metamorph and then tracked to a post-bleach image. The maximum value pixel is quantified. Apparent FRET efficiencies are calculated by:

as previously described $FRET_{eff} = \frac{(I_{post} - I_{pre})}{I_{post}}$ (**Verveer et al., 2006**).

## Transfections

Nups and LANS constructs were transfected 48 hr before imaging, with Fugene6 (Promega, Madison, WI). DN-Ran was transfected 16 hr before imaging. Cells were transfected with 1 µg DNA (for LANS and Nup133-mEGFP), 500 ng DNA (BFP-DNRan and Nup93-mEGFP), or 250 ng DNA (Nup54-mEGFP) and 3 µL FuGENE6 (Promega) in Opti-Mem I (Gibco) to a final volume of 100 µL according to the manufacturer's instructions.

## Digitonin permeabilization

Cells were permeabilized as previously described (**Adam et al., 1992**). Cells were incubated on ice for 5 min. Then cells were washed in cold transport buffer (TB) (20 mM HEPES, 110 mM potassium acetate, 5 mM sodium acetate, 2 mM magnesium acetate, and 1 mM ethylene glycol tetraacetic acid (EGTA) at pH 7.3). Cells were then incubated on ice in TB with 34 µg/mL digitonin (Sigma) for 5 min. Cells were then washed twice in cold TB and twice in 37C TB with 1.5% (wt/vol) 360kD polyvinylpyrrolidone (PVP, Sigma). Cells were imaged in 37C TB with 1.5% PVP. In order to confirm that 34 µg/mL digitonin left the nuclear envelope intact but resulted in the permeabilization of the plasma membrane, cells were incubated with 500 ng/mL R-phycoerythrin (ThermoFisher, Waltham, MA) for 10 min and then imaged. R-phycoerythrin is a 240 kD, so it should not readily diffuse through the NPC and can be used as a marker for nuclear integrity.

## Protein purification

His-tagged Ran in the pET28 vector (a gift from Dr. Günter Blobel) was transformed into BL21 (DE3) RIL-competent cells (Stratagene, St. Louis, MO). Expression was induced with 0.5 mM isopropyl β-D-1-thiogalactopyranoside (IPTG, Sigma) and cells were grown for 3 hr at 37C. Cells were spun at 6000 x g for 10 min at 4C (Sorvall SLC-6000, Waltham, MA) and the pellet was frozen overnight. The pellet was resuspended in 50 mM TRIS pH8, 150 mM NaCl with benzonase endonuclease (at 25 U/mL, Millipore, Waltham, MA) and rLysozyme (at 12 U/mL, Millipore) and 1x EDTA-free cOmplete protease inhibitors (Roche, Waltham, MA). The resuspended pellet was passed through a high-pressure homogenizer (Avestin EmuliFlex-C3, ATA Scientific, Taren Point, Australia) for lysis. The lysate was spun at 30,600 x g at 4C for 30 min (Sorvall SS-34 rotor). Imidazole was added to the lysate to a concentration of 10 mM. The supernatant was added to Ni-NTA beads (Qiagen, Germantown, MD) and nutated at 4C for 1.5–2 hr. This supernatant was then loaded on a column (Qiagen) and washed 6x with 50 mM Tris pH8, 150 mM NaCl, 20 mM imidazole. Ran was then eluted with 50 mM TRIS pH8, 150 mM NaCl, 300 mM imidazole in 500 µL fractions. The concentration of Ran was estimated by $OD_{600}$ and the fractions with the highest concentrations were pooled, buffer exchanged, and concentrated on Amicon Ultra Centrifugal Filter (10KD cutoff, Millipore). Samples were stored in aliquots of 10 mg in 50 mM TRIS pH 8, 150 mM NaCl, and 10% glycerol at −80C. Purification was confirmed by running the protein on a 4–12% Bis-Tris gel (Novex, Waltham, MA) and performing a Coomassie (PageBlue, Thermo Scientific).

His-tagged human Kap1α, Kap1β, and NTF2 in pET28 expression vectors were purified as described above with the following exceptions. These cultures were induced with 0.3 mM IPTG. Kap1α was grown overnight (~16 hr) at 18C following IPTG induction. Kap1α was stored in 200 μM aliquots, Kap1β in 20 μM aliquots, and NTF2 in aliquots of 11.4 mg/mL. The SV40 NLS was cloned into pNCStdTomato (a gift from Erik Rodriguez and Roger Tsien; Addgene plasmid #91767; http://n2t.net/addgene:91767; RRID:Addgene_91767). This protein was purified as above except it was grown overnight and no IPTG was added, because expression is constitutive.

## Cargo translocation

To determine whether purified Kaps were functional, a cargo translocation assay was performed. Cells were permeabilized as described above by treating with digitonin and subsequently removing cytosol by gentle washing. Cells were then incubated for 30 min at 37C with a transport mix. We tested two conditions: +Kaps and -Kaps. Both conditions contained the following base mix: 1.5 μM NLS-tdTomato, 0.1 mM GTP (ThermoFischer Scientific), 2 μM Ran, and 1 μM NTF2 in TB + pvp as described above. Both conditions also contained an ATP regenerating system, which includes 1 mM ATP (VWR), 1 mg/mL creatine phosphate (Sigma), and 15 U/mL creatine phosphokinase (Millipore). Finally, the two conditions either did or did not contain a receptor mix (+/- Kaps), consisting of 1.5 μM Kap1α and 1 μM Kap1β.

After the incubation period, we subsequently washed the cells to remove excess cargo and transport elements from the nuclear periphery. Cells were incubated with live-cell Hoechst (NucBlue Live ReadyProbes, Hoeschst 33342, ThermoFischer Scientific). Cells were then imaged to quantify the nuclear NLS-tdTomato. All excitations were done in a 'semi'-TIRFM excitation mode, which restricts fluorescence to a few micrometers near to the coverslip to illuminate more of the nucleoplasm. Nuclear regions were determined by constructing a mask in FIJI using the Hoescht staining and finding the average intensity in arbitrary units of the tdTomato signal (*Schindelin et al., 2012*).

## Ran-GTP loading and karyopherin removal

Ran was loaded with GDP using published methods (*Lowe et al., 2015*). 1 mM Ran was incubated with 50 mM GDP in 10 mM HEPES pH 7.3, 100 mM NaCl and 10 mM EDTA at RT for 30 min. Then the sample was diluted 2.5x in four steps at 1 min intervals in 10 mM HEPES pH 7.3, 100 mM NaCl, 10 mM EDTA and $MgCl_2$ such at the final concentration of $MgCl_2$ was 25 mM. The solution was then dialyzed in TB overnight.

To remove kaps, cells were treated with Ran mix using published methods (*Adam et al., 1992*; *Kapinos et al., 2017*). Cells were incubated with Ran mix in TB+pvp for 1 hr (2 mM GTP, 0.1 mM ATP, 4 mM creatine phosphate, 20 U/mL creatine kinase, 5 M RanGDP, 4 μM NTF2, and 1 mM DTT).

## CRISPR cell lines

Guides and repair templates were constructed by using the Benchling CRISPR design tool (Benchling, San Francisco, CA). Guides were cloned into the pSpCas9(BB)−2A-Puro (PX459) V2.0 plasmid following standard protocols as developed by the Zhang lab (*Ran et al., 2013*). The repair templates for Nup54[494]-mEGFP(0) and Nup133-mEGFP(−9) were synthesized by GenScript as dsDNA. This DNA was then A'-tailed and placed in a TOPO-TA Cloning Vector (2.1-TOPO, ThermoFisher Scientific). Repair templates for Nup54[494]-mEGFP(1) and Nup54[494]-mEGFP(2) and Nup133-mEGFP(−8) were cloned from the repair template plasmids constructed above via the QuikChange Lightning Site-Directed Mutagenesis Kit (Agilent, Santa Clara, CA). For each CRISPR cell line, cells were transfected with linearized repair templates and the PX459 plasmid containing the appropriate guides using FuGENE HD (Promega) with a ratio of Reagent : DNA of 3 : 1 according to manufacturer's protocol. Transfected cells were selected via antibiotic selection with puromycin (0.6 μg/mL, InvivoGen, San Diego, CA) for 48 hr. Cells were released from antibiotic selection and allowed to recover for 24 hr. GFP positive cells were collected in the Rockefeller University Flow Cytometry Resource Center using a FACSAria II flow cytometer (BD Biosciences, San Jose, CA). Cells were resuspended in PBS (no calcium/no magnesium, Gibco), 0.5% (vol/vol) bovine serum albumin (BSA, Sigma), 5 mM EDTA (Gibco), and 15 mM HEPES (ThermoFisher). GFP positive cells were sorted and screened via live-cell microscopy for signal at the nuclear rim (~95% of GFP positive cells showed nuclear rim GFP signal). Cell lines were screened for homozygosity via PCR. Two bands were seen for heterozygotes and one

band for homozygotes, both fragments were sequenced to confirm amplification of correct region and proper incorporation of the mEGFP. In order to confirm that the mEGFP was not incorporated elsewhere and producing another labeled protein within our cells, western blots were performed. Protein samples were run on a 4–12% Bis-Tris gel (Novex) and Western blots were performed with the following antibodies: α-GFP (Clontech Living Colors 632381 (JL-8); RRID:AB_2313808, mouse monoclonal, 1:1,000) and goat α-mouse–HRP (Sigma–Aldrich; RRID:AB_258476, 1:50,000). Membranes were then striped with Restore Western Blot Stripping Buffer (ThermoFisher) and western blots were performed with the following antibodies: α-actin (Abcam; RRID:AB_476743, 1:1,000) and goat α-mouse–HRP (Sigma–Aldrich; RRID:AB_258476, 1:50,000). All blocking and incubations were done in 5% (wt/vol) nonfat milk powder in Tris-buffered saline with Tween 20.

## Quantification and statistical analysis

Statistical analyses were performed using MATLAB Version 2019b and are described in the figure legends and in the Method Details.

## Acknowledgements

We thank Günter Blobel, Elias Coutavas, and Ivo Melčák for Nup58, Kap-Alpha1, and Kap-Beta1 cDNA and for the his-Ran plasmid. pSpCas9(BB)−2A-Puro (PX459) V2.0 was a gift from Feng Zhang (Addgene plasmid # 62988). DN-RanQ69L was a gift from Jay Brenman (Addgene plasmid # 30309) and pTriEx-mCherry::LANS4 was a gift from Brian Kuhlman (Addgene plasmid # 60785). We are grateful for the assistance of the Flow Cytometry Resource Center at The Rockefeller University. We thank Günter Blobel, Philip Coffino and Elias Coutavas for helpful discussions. Funding: The Howard Hughes Medical Institute Gilliam Fellowship (J P).

## Additional information

### Funding

| Funder | Grant reference number | Author |
| --- | --- | --- |
| Howard Hughes Medical Institute | Gilliam Fellowship | Joan Pulupa |

The funders had no role in study design, data collection and interpretation, or the decision to submit the work for publication.

### Author contributions

Joan Pulupa, Conceptualization, Data curation, Software, Formal analysis, Validation, Investigation, Visualization, Methodology, Writing - original draft, Project administration, Writing - review and editing; Harriet Prior, Data curation, Validation, Investigation, Methodology, Writing - review and editing; Daniel S Johnson, Validation, Methodology, Writing - review and editing; Sanford M Simon, Conceptualization, Resources, Supervision, Funding acquisition, Project administration, Writing - review and editing

### Author ORCIDs

Joan Pulupa (ID) https://orcid.org/0000-0003-3858-1886
Daniel S Johnson (ID) http://orcid.org/0000-0001-8906-0509
Sanford M Simon (ID) https://orcid.org/0000-0002-8615-4224

### Decision letter and Author response

Decision letter https://doi.org/10.7554/eLife.60654.sa1
Author response https://doi.org/10.7554/eLife.60654.sa2

# Additional files

## Supplementary files

- Source code 1. Analysis code for quantification of orientation.

- Supplementary file 1. Table of Nup-mEGFP transfected fusion protein linkage identities. The size of mEGFP deletion describes the number of amino acids deleted from the amino terminus of mEGFP and the net linker size describes the number of amino acids in the linker minus the deletions from the Nup and mEGFP.

- Supplementary file 2. Table of Nup-mEGFP cell line linkage identities. The size of mEGFP deletion describes the number of amino acids deleted from the amino terminus of mEGFP and the net linker size describes the number of amino acids in the linker minus the deletions from the Nup and mEGFP.

- Transparent reporting form

## Data availability

All data generated or analysed during this study are included in the manuscript and supporting files. Source data files have been provided for all data figures. Imaging data has been uploaded to figshare (https://figshare.com/projects/Conformation_of_the_nuclear_pore_in_living_cells_is_modulated_by_transport_state/93755).

The following dataset was generated:

| Author(s) | Year | Dataset title | Dataset URL | Database and Identifier |
|-----------|------|---------------|-------------|-------------------------|
| Pulupa J, Prior H, Johnson DS, Simon SM | 2020 | Conformation of the nuclear pore in living cells is modulated by transport state | https://figshare.com/projects/Conformation_of_the_nuclear_pore_in_living_cells_is_modulated_by_transport_state/93755 | figshare, 93755 |

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
