## [Decision Letter]

**Acceptance summary:**

Experimental approaches capable of revealing the conformations of protein assemblies in living cells are likely to provide new insights into a host of biological questions. Here, the authors combine optical methods to develop just such a technique. This work leverages this new approach to reveal how the transport state of individual nuclear pore complexes is coupled to changes in its conformation, highlighting the flexibility of this structure.

**Decision letter after peer review:**

Thank you for submitting your article "Cargo modulates the conformation of the nuclear pore in living cells" for consideration by *eLife*. Your article has been reviewed by three peer reviewers, one of whom is a member of our Board of Reviewing Editors, and the evaluation has been overseen Vivek Malhotra as the Senior Editor. The reviewers have opted to remain anonymous.

The reviewers have discussed the reviews with one another and the Reviewing Editor has drafted this decision to help you prepare a revised submission.

Summary:

In this manuscript Pulupa et al. develop the application of "pol-TIRFM" applied to individual nuclear pore complexes in living cells. Technically, this work builds on their past published papers (Biophys J. 2013, Nat Struct Mol Biol. 2011, Biophys J. 2010 Sep 22;99(6):1706-17), now extended to single NPC resolution. After exploring suitable fluorescent protein fusions of nucleoporins in the Y-complex (Nup133), and two aspects of the inner ring (Nup93 and the FG nups, Nup58 and Nup54) that are sensitive to the orientation of the fluorophore, they then examine how different perturbations to transport affect these orientation sensors. Numerous treatments expected to alter nucleocytoplasmic transport change the orientation of Nup54 and Nup58, including starvation, inhibition of Crm1-mediated nuclear export, and expression of a dominant-negative form of the Ran-GTPase. Another observation is that permeabilizing cells alters the orientation of Nup54 (mimicking the perturbations listed above), but this can be restored by incubation with Kapa1 and Kapb. Consistent with the interpretation that the Kaps alter the conformation of Nup54, the orientation change upon permeabilization can be augmented (and sped up) by addition of Ran-GTP, which was previously shown to strip Kaps from the NPC. Last, the authors use an orthogonal approach (a FRET reporter in Nup62); the Nup62 protomers show increased FRET upon starvation.

Overall there was enthusiasm for the approach, particularly its potential to provide a quantitative analysis of conformational changes in the NPC, as well as an appreciation for the technical feat of detecting anisotropy changes on the single NPC level. In terms of the biology, while there is a rich understanding of the architecture of the NPC from structural biology, a major unanswered question is the dynamic conformations of the nuclear pore complex that are sampled, under what conditions such structural changes occur, and the physiological relevance. In this respect, the reviewers found the evidence that the reporters reflect changes in the inner ring of the NPC in response to starvation and other perturbations to be largely convincing. There were, however, several aspects of the work that the reviewers felt could be improved. A particular concern was how the orientation changes relate to the structural states of the NPC. A second was on the ability to fully interpret the changes in the orientation of Nup54 and Nup58 with regards to large scale changes in the NPC. There were also several areas that the reviewers felt could use further clarification, as outlined below. Lastly, the reviewers expect that the authors will want to comment on their results in the context of the recent work from the Beck lab on starvation and NPC structure.

Essential revisions:

1) There was a consensus that getting beyond a "yes/no" of whether a change occurred to some concept of the kind of orientation change that took place in the context of the NPC would greatly strengthen the work. As presented, it was not clear how much Nup orientation changed nor how this relates to the state of the NPC itself (e.g., dilated vs. constricted) or its engagement with components like transport factors. The reviewers made several suggestions for how this might be achieved. First, the reviewers felt that the authors should address/interpret/discuss the quantitative differences/similarities in the p:s behavior across test conditions, particularly when this could be put into greater context of the NPC structure. Another approach that the reviewers considered is whether the authors can better leverage the extensive prior structural studies on the NPC and nucleoporins, the known orientation of the NPC relative to the evanescent wave and the polarization in this experimental set-up, and the differences in the observed p:s ratio upon stepping the linker by single amino acids, to interpret more detail about how the perturbations are influencing the position of the tagged nups in the NPC. Combining such data with a model-based theory could constrain the possibilities. Last, and related to the prior two points, was the possibility that the new work from the Beck lab might better position the authors to bridge their observations with more specific changes in the NPC (see next point).

2) While this paper has been under review a related paper (albeit in a different model) from Martin Beck's group (Allegretti et al.) was posted to the bioRxiv. The reviewers felt it important that a revision address this work in their interpretation and discussion. Specifically: 1) How do the observations compare about the response to starvation (recognizing that the fission yeast and mammalian NPC have different structure)?; 2) Can the authors draw any parallels with the Allegretti paper, particularly the increase of FRET in Nup62 after starvation? Can the authors suggest if this implies constriction or dilation?; 3) the Beck paper suggests that the observed changes are distinct from those proposed in the "Ring Cycle" – this should be taken into account in the Discussion here.

3) The reviewers felt it important to discriminate between the effects of Kap binding to the FG nucleoporins in the act of transporting cargo versus effects solely of Kap binding (independent of transport). This point could be addressed using Kap separation-of-function mutants such as the N-terminal truncation of Kap-β.

4) There were several questions related to the observation that the specific nucleoporins that showed changes in orientation in response to transport properties of the NPC directly associate with karyopherins through their FG-domains, which are part of the same polypeptide with the structured domains (where the fluorophore is fused). Thus, the major movements could be restricted to within the same polypeptide. This begs the question of whether transport alters the inner ring in a concerted manner beyond the state of the individual FG nucleoporin. One suggestion is to monitor, in the transient transfections, Nup54 truncations lacking the FG-repeat domain still respond in the same manner, which would be strong evidence propagation of the inner ring complex independent of engagement of its FG region. Ideally this would be achieved by titrating in the FG-less Nup54.

5) Further explanation/discussion is needed for the observed effects of adding Ran mutant/leptomycin B. Specifically, when hydrolysis is blocked, RanQ69L-GTP-importin complexes should not dissociate. If so, importins would no longer be available for undertaking another round of import in the cytoplasm. Likewise, NES-cargo-RanQ69L-GTP-exportin triple complexes should not disassemble. In this event, transiting pools of RanQ69L-GTP-importin and NES-cargo-RanQ69L-GTP-exportin ought to remain at NPCs due to binding to the FG Nups. How is it the p:s ratio in Figure 4L shares a similar trend with Figure 4H (i.e., a decrease)? How might this compare to what happens in the NPC after permeabilization (Figure 4H)? Leptomycin B inhibits NES-cargo binding to Crm1, thereby decreasing the amount of NES-cargo being exported through the NPC. Can the authors please attempt an explanation as to how the p:s ratio in Figure 4J shares a similar trend to Figure 4H and L (i.e., a decrease)?

6) Can the authors please comment on the difference in observed behavior reported in Figure 5F/H and Figure 4H? The Nup54 p:s ratio increases upon removing endogenous karyopherins by +Ran mix (pink, Figure 5F). Adding exogenous karyopherins then returns the p:s ratio towards pre-Ran mix values (blue). Moreover, no change to the p:s ratio is seen when Ran mix is absent (Figure 5H). Again, Nup54 p:s ratio decreases "in a buffer containing no transport factors or cargo" (Figure 4H) when these experiments do not seem so different. What might bring about these differences?

7) The reviewers were concerned that some statements in the manuscript were either over-sold or over-interpreted. Specifically, please revise to avoid "first time statements" including: "we have developed the first tools to directly observe the orientational dynamics of Nups within a single NPC in living cells and to track these dynamics over time 1 with different cargo conditions", which is dismissive of the various prior studies in the literature showing that the NPC can change its structure. While the authors employ live-cells (clearly an advance), their "dynamics" are in reality snapshots at steady state, at least several minutes or hours apart. In addition, the reviewers would ask the authors to revise the manuscript to acknowledge that they cannot rule out that a conformational change occurs outside the inner ring because it could not be observed using this approach. Lastly, the reviewers suggest that the title "Cargo modulates the conformation of the nuclear pore in living cells" is overstated. Suggested: "Transport state modulates the conformation of the nuclear pore in living cells"

8) The reviewers would like the authors to report not just orientation (the p:s ratio in the manuscript) but also rigidity – that is, the anisotropy. This was seen as particularly important because it is unclear whether rigidity of inner ring Nups will vary with nuclear transport state.

---

## [Author Response]

Essential revisions:1) There was a consensus that getting beyond a "yes/no" of whether a change occurred to some concept of the kind of orientation change that took place in the context of the NPC would greatly strengthen the work. As presented, it was not clear how much Nup orientation changed nor how this relates to the state of the NPC itself (e.g., dilated vs. constricted) or its engagement with components like transport factors. The reviewers made several suggestions for how this might be achieved. First, the reviewers felt that the authors should address/interpret/discuss the quantitative differences/similarities in the p:s behavior across test conditions, particularly when this could be put into greater context of the NPC structure.

Thank you for this suggestion. We are strong believers in doing quantitative analysis of the results, especially across different test conditions. We have analyzed the quantitative differences across different treatments of the same probe. However, we are also cautious about over-interpreting our results. In the absence of additional structural information (see below), we think it is inappropriate to try to calibrate the physical magnitude of the change of the orientation of the Nup.

Therefore, we have added further interpretation to both the analysis of Figure 4:

“The effects of starvation and permeabilization appear to be more pronounced than that of the drug treatment and the dominant-negative Ran. This may be due to a more severe reduction in cargo flux, and therefore a more complete assumption of the NPC of the state adopted when cargo flux is reduced.”

Another approach that the reviewers considered is whether the authors can better leverage the extensive prior structural studies on the NPC and nucleoporins, the known orientation of the NPC relative to the evanescent wave and the polarization in this experimental set-up, and the differences in the observed p:s ratio upon stepping the linker by single amino acids, to interpret more detail about how the perturbations are influencing the position of the tagged nups in the NPC. Combining such data with a model-based theory could constrain the possibilities.

We appreciate this suggestion! In the past, we have used fluorescence polarization measurements to determine the orientation of Nup133 and the y-shaped complex within the NPC. We did this by comparing results of measurements of anisotropy to computational predictions based on the crystal structure of Nup133 and the cryo-EM structure of the y-shaped complex. These experiments were easier to perform with Nup133 because we were simply testing how the y-shaped complex fit into the NPC, and there was much established structural knowledge of the y-shaped complex.

We have been intrigued by the possibility of combining the orientational changes we are able to measure with structural data of the NPC. We believe that this is a potential next step. Determining the precise angle of the change of orientation of the inner ring nups would require additional structural information about how exactly these alpha helices are incorporated into the NPC, as well as considerable computational work. Since August 2019 we have been speaking with Jan Kosinski about joining forces to use his structural measurements with our fluorescence. Alas, the structural information is still not resolved enough to join these data sets so this kind work will have to be outside of the scope of this project. We look forward to building, in the future, a computational model to calculate the angle changes of the alpha helices to which our probes are conjugated.

Last, and related to the prior two points, was the possibility that the new work from the Beck lab might better position the authors to bridge their observations with more specific changes in the NPC (see next point).2) While this paper has been under review a related paper (albeit in a different model) from Martin Beck's group (Allegretti et al.) was posted to the bioRxiv. The reviewers felt it important that a revision address this work in their interpretation and discussion.

Thank you. We have added discussion of this paper to our manuscript. This beautiful work with an orthogonal technique is an independent confirmation of our data showing conformational changes in the main scaffold of the NPD that are associated with transport state. We started an active discussion with Kosinski and Beck over a year ago, once we felt we had a robust, reproducible story. We have been trying to coordinate with them both the kinds of perturbations on transport that we use as well as the Nups we focus on. However, we want to be cautious about not overinterpreting our data. We are confident in the measurement of the change of FRET and the change in the relative excitation by p and s light. Consequently, we are confident in changes of orientation that correlate with the change of transport. However, we feel as if it could be a disservice to the reader to overinterpret the results at this point in terms of a specific model.

Specifically: 1) How do the observations compare about the response to starvation (recognizing that the fission yeast and mammalian NPC have different structure)?;

We have added the following section to the Discussion:

“While this manuscript was in review, cryo-EM tomograms revealed in situ NPCs taken from *S. cerevisiae* cells in exponential growth phase were ~20 nm larger in diameter than isolated NPCs from *S. cerevisiae*, underscoring the potential flexibility of the NPC (Allegretti et al., 2020).“

And revised our analysis as follows:

“The conformational changes we observe in the NPCs of living cells may be coincident with cargo translocation or may be an indication that the scaffold of the NPC serves as a dynamic gate that can regulate nuclear trafficking. […] Further studies will be needed to determine the exact conformations we are monitoring in this study, and to what extent the conformations we observe align with these models. ”

2) Can the authors draw any parallels with the Allegretti paper, particularly the increase of FRET in Nup62 after starvation? Can the authors suggest if this implies constriction or dilation?;

Yes, we have added the following to the Discussion:

In addition to changes in orientation, we also observe increases in FRET in the FG-regions of Nup62 upon starvation, which is also consistent with a spatial rearrangement and constriction of the NPC diameter.

3) the Beck paper suggests that the observed changes are distinct from those proposed in the "Ring Cycle" – this should be taken into account in the Discussion here.

We have made the above changes to the Discussion to explain the difference.

3) The reviewers felt it important to discriminate between the effects of Kap binding to the FG nucleoporins in the act of transporting cargo versus effects solely of Kap binding (independent of transport). This point could be addressed using Kap separation-of-function mutants such as the N-terminal truncation of Kap-β.

We are very interested in distinguishing between the effect of Kaps binding on their own from Kaps transporting cargo (including specific cargo such as large cargo or inner nuclear membrane proteins).

In order to test the effect of Kaps devoid of cargo on the NPC, we introduced karyopherins alone in Figure 5H. Although this is not the Kap separation-of-function mutant, there was no additional cargo introduced in this experiment. In the absence of cargo, we did observe that the Kaps alone caused the NPC to be in an alternate conformation. In the future, we hope to track individual cargos transiting through the pore while monitoring the orientation of Nups to see the how transiting cargo affects the conformation of the NPC.

4) There were several questions related to the observation that the specific nucleoporins that showed changes in orientation in response to transport properties of the NPC directly associate with karyopherins through their FG-domains, which are part of the same polypeptide with the structured domains (where the fluorophore is fused). Thus, the major movements could be restricted to within the same polypeptide. This begs the question of whether transport alters the inner ring in a concerted manner beyond the state of the individual FG nucleoporin. One suggestion is to monitor, in the transient transfections, Nup54 truncations lacking the FG-repeat domain still respond in the same manner, which would be strong evidence propagation of the inner ring complex independent of engagement of its FG region. Ideally this would be achieved by titrating in the FG-less Nup54.

Thank you for this suggestion. We have performed this experiment, and we show that Nup54 orientational sensors lacking an FG-domain experience the same conformational changes as Nup54 orientational sensors containing a wild-type FG-domain. We appreciate the suggestion and agree that these results provide evidence that the conformational change is propagated throughout the inner ring, independent of the engagement karyopherins with individual Nup whose orientation is being altered.

We have added this figure to our manuscript as Figure 3—figure supplement 2. We have also added the following statement to our Results:

“The orientation shift of Nup54 was also seen in Nup54 orientational sensors where the FG-Nup domain was eliminated. This result suggests that these orientational changes are propagated throughout the inner ring, independent of whether the individual Nup54 polypeptide containing orientational sensor is bound to a kap.”

5) Further explanation/discussion is needed for the observed effects of adding Ran mutant/leptomycin B. Specifically, when hydrolysis is blocked, RanQ69L-GTP-importin complexes should not dissociate. If so, importins would no longer be available for undertaking another round of import in the cytoplasm. Likewise, NES-cargo-RanQ69L-GTP-exportin triple complexes should not disassemble. In this event, transiting pools of RanQ69L-GTP-importin and NES-cargo-RanQ69L-GTP-exportin ought to remain at NPCs due to binding to the FG Nups. How is it the p:s ratio in Figure 4L shares a similar trend with Figure 4H (i.e., a decrease)? How might this compare to what happens in the NPC after permeabilization (Figure 4H)? Leptomycin B inhibits NES-cargo binding to Crm1, thereby decreasing the amount of NES-cargo being exported through the NPC. Can the authors please attempt an explanation as to how the p:s ratio in Figure 4J shares a similar trend to Figure 4H and L (i.e., a decrease)?

All methods of decreasing cargo flux through the NPC resulted in the same orientational change in Nup54. (That is that 4F, 4H, 4J, and 4L) all showed the same trend. One possible explanation that would explain this shift is that in all of these cases, the cargo flux is decreased. It is possible that the conformational plasticity we observe is necessary for the transport of large cargoes or inner nuclear membrane proteins, transport of all of which would be reduced in cells under starvation, permeabilization, leptomycin B, or dominant-negative ran treatment. Thus, the conformational change is permissive for transport, but is neutral as to the direction of transport. Specifically, in the text we state:

“These results suggest that attenuating the transport of NLS-driven cargo shifts the orientation of select alpha-helical domains of inner ring Nups. The direction of the p:s ratio shift of Nup54-mEGFP494 is consistent among all mechanisms of reducing cargo flux.”

6) Can the authors please comment on the difference in observed behavior reported in Figure 5F/H and Figure 4H? The Nup54 p:s ratio increases upon removing endogenous karyopherins by +Ran mix (pink, Figure 5F). Adding exogenous karyopherins then returns the p:s ratio towards pre-Ran mix values (blue). Moreover, no change to the p:s ratio is seen when Ran mix is absent (Figure 5H). Again, Nup54 p:s ratio decreases "in a buffer containing no transport factors or cargo" (Figure 4H) when these experiments do not seem so different. What might bring about these differences?

We apologize for the confusion and we have sought to clarify these points in text. We see a change in the orientation of Nup54 upon permeabilization of the cells. This, we believe is the result of the reduction of karyopherins and cargo. In the case of Figure 4H, the control (CM, left) is an unpermeabilized cell in complete media. The experiment (PERM, right) is a permeabilized cell. Thus, the right column of Figure 4G is equivalent to the t=0 (left most column) of Figure 5F-I.

We have modified text when describing Figure 4H and Figure 4G to help clarify this point:

“When we permeabilize cells and incubate in transport buffer plus 1.5% (wt/vol) 360kD polyvinylpyrrolidone to mimic cytosolic conditions, we observed a distinct shift in the orientation of Nup54^494^ compared to the orientation of Nup54^494^ in unpermeabilized cells grown in complete media. We saw no change in the orientation of Nup133 in permeabilized cells compared to Nup133 in unpermeabilized cells grown in complete media (Figure 4G-H).”

7) The reviewers were concerned that some statements in the manuscript were either over-sold or over-interpreted. Specifically, please revise to avoid "first time statements" including: "we have developed the first tools to directly observe the orientational dynamics of Nups within a single NPC in living cells and to track these dynamics over time 1 with different cargo conditions", which is dismissive of the various prior studies in the literature showing that the NPC can change its structure. While the authors employ live-cells (clearly an advance), their "dynamics" are in reality snapshots at steady state, at least several minutes or hours apart. In addition, the reviewers would ask the authors to revise the manuscript to acknowledge that they cannot rule out that a conformational change occurs outside the inner ring because it could not be observed using this approach. Lastly, the reviewers suggest that the title "Cargo modulates the conformation of the nuclear pore in living cells" is overstated. Suggested: "Transport state modulates the conformation of the nuclear pore in living cells"

Thank you for this response. We agree with these comments and have made the suggested changes in text and in the title. In addition to eliminating “first time statements” and replacing the word “dynamics,” we also added the following statement to our Discussion:

“It is possible that those regions are moving translationally and thus escaping our detection, which is sensitive only to orientation changes with respect to the nuclear-cytoplasmic axis.”

8) The reviewers would like the authors to report not just orientation (the p:s ratio in the manuscript) but also rigidity – that is, the anisotropy. This was seen as particularly important because it is unclear whether rigidity of inner ring Nups will vary with nuclear transport state.

We have been struggling to report anisotropy values using pol-TIRF. However, we do not yet feel comfortable reporting absolute values for two reasons. First, anisotropy will always be a convoluted measurement, containing information about both orientation and rigidity. In a system where the Nups are organized with dependence around an optical axis, it is highly challenging to separate the two. Another obstacle to reporting anisotropy is the mixing of light of different polarizations when they are collected using a high-NA TIRF objective [in a nutshell, at the higher angles, a component of the light polarized in z is split to the x and y axis. This extent of this varies as one moves more to the periphery of the lens.]. Thus, as the light is collected, parallel and perpendicular emissions are mixed, and at the high-NA we are using (1.49), the correction factors for that have been developed break down. We have been working during this response to address this issue but we realize that a full reworking with proper calibrations and validations of these correction factors for lenses of high numerical aperture is its own two year project. Although, we are interested in developing methods to measure anisotropy in TIRF, we do not believe we currently are able to report anisotropy in a rigorous manner.